# Transcriptome dynamics of the *Myxococcus xanthus* multicellular developmental program

José Muñoz-Dorado[1]*, Aurelio Moraleda-Muñoz[1], Francisco Javier Marcos-Torres[1], Francisco Javier Contreras-Moreno[1], Ana Belen Martin-Cuadrado[2], Jared M Schrader[3], Penelope I Higgs[3], Juana Pérez[1]

[1]Departamento de Microbiología, Facultad de Ciencias, Universidad de Granada, Granada, Spain; [2]Departamento de Fisiología, Genética y Microbiología, Universidad de Alicante, Alicante, Spain; [3]Department of Biological Sciences, Wayne State University, Detroit, United States

**Abstract** The bacterium *Myxococcus xanthus* exhibits a complex multicellular life cycle. In the presence of nutrients, cells prey cooperatively. Upon starvation, they enter a developmental cycle wherein cells aggregate to produce macroscopic fruiting bodies filled with resistant myxospores. We used RNA-Seq technology to examine the transcriptome of the 96 hr developmental program. These data revealed that 1415 genes were sequentially expressed in 10 discrete modules, with expression peaking during aggregation, in the transition from aggregation to sporulation, or during sporulation. Analysis of genes expressed at each specific time point provided insights as to how starving cells obtain energy and precursors necessary for assembly of fruiting bodies and into developmental production of secondary metabolites. This study offers the first global view of developmental transcriptional profiles and provides important tools and resources for future studies.

DOI: https://doi.org/10.7554/eLife.50374.001

*For correspondence:
jdorado@ugr.es

**Competing interests:** The authors declare that no competing interests exist.

## Introduction

*Myxococcus xanthus* is a soil-dwelling δ-proteobacterium that exhibits a complex multicellular life cycle with two phases: growth and starvation-induced development (*Muñoz-Dorado et al., 2016*). When nutrients are available, cells divide to produce a community known as swarm. Swarms are predatory (although not obligate) and can digest prokaryotic and eukaryotic microorganisms (*Pérez et al., 2016*). Upon starvation, cells in the swarm enter a developmental program, during which they migrate into aggregation centers and climb on top of each other to build macroscopic structures termed fruiting bodies. To form fruiting bodies, starving cells glide on solid surfaces by using two motility systems, known as A- (adventurous) and S- (social) motility, which allow individual cell movement or group movement that requires cell-cell contact, respectively (*Mauriello et al., 2010*; *Nan et al., 2014*; *Islam and Mignot, 2015*; *Chang et al., 2016*; *Schumacher and Søgaard-Andersen, 2017*). After completion of aggregation (24 hr post-starvation), cells differentiate into environmentally resistant myxospores, which are embedded in a complex extracellular matrix (*Figure 1*). Each fruiting body contains ≈$10^5$–$10^6$ myxospores. Interestingly, only ≈10% of the starving population become myxospores (*O'Connor and Zusman, 1991a*) as most cells (around 60%) undergo programmed cell death, most likely to provide the rest of the population enough nutrients to successfully build fruiting bodies (*Wireman and Dworkin, 1977*; *Nariya and Inouye, 2008*). The remaining cells differentiate into a persister-like state, termed peripheral rods (PR), which surround the fruiting bodies (*O'Connor and Zusman, 1991a*; *O'Connor and Zusman, 1991b*; *O'Connor and*

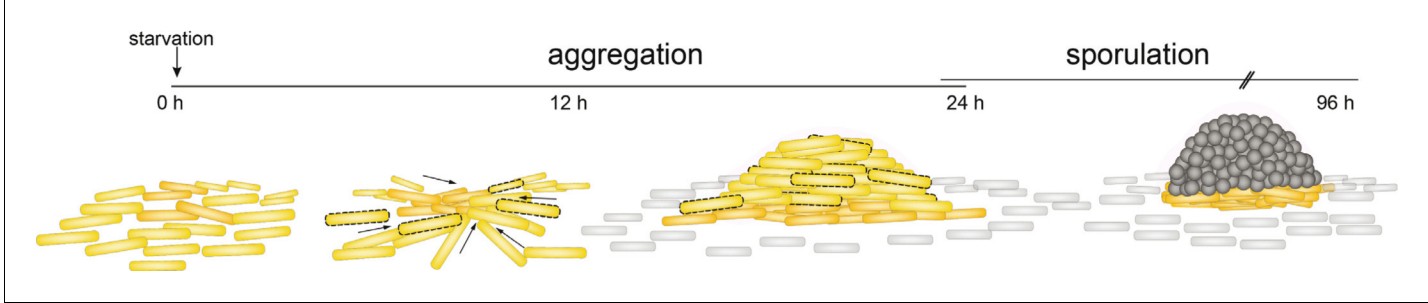

**Figure 1.** Schematic of the *M. xanthus* developmental program. The time line indicates aggregation and sporulation phases. *M. xanthus* cells (yellow rods) aggregate into mounds (arrows indicate gliding to aggregation centers) and then differentiate into resistant spores (gray circles) to produce mature fruiting bodies. Peripheral rods (gray rods) remain outside of the fruiting bodies as a distinct differentiated state. Cells undergoing lysis are depicted with dashed lines.

DOI: https://doi.org/10.7554/eLife.50374.002

*Zusman, 1991c*). While PRs are morphologically similar to vegetative cells, myxospores are coccoid and are surrounded by a thick coat mainly consisting of polysaccharides (*Kottel et al., 1975*; *Müller et al., 2010*; *Müller et al., 2012*; *Holkenbrink et al., 2014*). Myxospores can germinate when nutrients are available, and collective germination of myxospores from a fruiting body generates a small swarm that facilitates cooperative feeding.

The developmental program is directed by sophisticated, but not completely defined, genetic regulatory networks, which are coupled to a series of intra- and extra-cellular cues (*Kroos, 2017*; *Muñoz-Dorado et al., 2016*). The first cue is starvation, which triggers accumulation of cyclic-di-GMP and, via the stringent response, guanosine penta- and tetraphosphate [(p)ppGpp] inside the cells. These global signals somehow activate four master cascade modules (Nla24, Mrp, FruA, and bacterial enhancer-binding proteins [bEBPs]), which interconnect to control the correct timing of gene expression (*Kroos, 2017*). Proper progression of development requires intercellular communication, wherein cells produce and transmit five sequential extracellular signals, named A, B, C, D, and E (*Bretl and Kirby, 2016*).

Although much knowledge has been generated in the last 40 years about the *M. xanthus* developmental cycle, especially with respect to signaling and gene regulatory networks, we are far from having an overall picture of all the events that occur during aggregation and sporulation. Several partial transcriptome analyses from developmental samples based on microarrays have been published, which were restricted to a few genes related to bEBPs (*Jakobsen et al., 2004*; *Diodati et al., 2008*), two-component systems (TCSs) (*Shi et al., 2008*), A-signaling genes (*Konovalova et al., 2012*), or lipid metabolism (*Bhat et al., 2014*). Here, we used RNA-Seq technology to measure global changes in transcript abundance at seven time points during *M. xanthus* development, which represents a substantial step forward compared to previous analyses. We found that at least 19.6% of *M. xanthus* genes (1415/7229) had statistically significant changes in transcript abundance during development. These data and analyses provide, for the first time, a comprehensive view of the transcriptional regulatory patterns that drive the multicellular developmental program of this myxobacterium, offering an essential scaffold for future investigations.

## Results and discussion

### Transcriptome analysis of the developmental program by RNA-Seq

Global gene expression patterns were examined by RNA-Seq analysis of the wild-type *M. xanthus* strain DK1622 developed on nutrient limited CF agar plates. RNA was harvested from two independent biological replicates at 0, 6, 12, 24, 48, 72, and 96 hr of development, reverse transcribed to cDNA, and sequenced by Illumina methodology. On average, 54.72 million read pairs and a coverage of 591X was obtained. After removing the ribosomal sequences, the genome coverage varied from 5.52 to 14.18X (median of 10.49X), enough to provide an adequate coverage of the mRNA fraction. The two sample-replicates showed a high degree of concordance in gene expression ($R^2$

correlation >0.98), with the exception of 24 hr samples ($R^2$correlation = 0.80), which may be related to lack of synchrony between cells in the transition from aggregation to sporulation. The median of both values was utilized for further analysis (*Table 1* and *Table 1—source data 1*).

As a first data validation step, we determined the expression profiles of two different developmental genes using β-galactosidase transcriptional reporters [*spiA*::Tn5-*lacZ* (strain DK4322) and *fmgE*::Tn5-*lacZ* (strain DK4294)] (*Kroos et al., 1986*) from cells developed under the same conditions used in this study. Comparison of these β-galactosidase activities to the RNA-Seq data indicated the patterns were similar and in agreement with those obtained with other strategies (*Figure 2*). Moreover, the expression profiles of many genes that have been previously characterized from β-galactosidase reporter activity, qRT-PCR, or microarray analyses were compared with those obtained with our data. This analysis revealed a general agreement (*Figure 2—source data 1*).

## Gene expression profiles organize into 10 developmental groups (DGs)

To identify developmentally regulated transcripts with similar expression patterns, genes containing measured RPKM (reads per kilobase pair of transcript per million mapped reads) values for all time points were further analyzed. First, all genes with <50 reads and/or high replicate variability between the two replicate datasets ($R^2$ correlation <0.7) were removed (*Table 1—source data 2*). 1557/7229 (21.5 %) genes passed quality criteria filters. Some of these genes (142; 9.1%) were not significantly (>2 fold) up- or down regulated during the developmental program, suggesting these genes could be considered constitutively expressed. 1415 significantly regulated genes (90.9% of the genes passing quality control) were then analyzed for clusters of similar expression patterns. Briefly, initial evaluation of several clustering methods (see Materials and methods for details) revealed kmeans clustering with 6–12 clusters produced the best clustering of genes with similar expression profiles. Refinement of kmeans clusters by visual inspection indicated 10 DGs best explained the number of

**Table 1.** Statistical analysis of the *M. xanthus* DK1622 transcriptome raw data.
Data for each of the replicas at 0, 6, 12, 24, 48, 72 and 96 hr of development are shown.

| Sample name | #Gb | #mapped reads | #rRNA-reads | #clean reads (Non-rRNA) | %rRNA rate | Coverage (x) | $R^2$ correlation |
|---|---|---|---|---|---|---|---|
| WT_0_1 | 5.70 | 61906718 | 60713758 | 1192960 | 98.07 | 13.05 | 0.99 |
| WT_0_2 | 5.62 | 61117544 | 59826984 | 1290560 | 97.89 | 14.12 | |
| WT_6_1 | 5.02 | 54468281 | 53756387 | 711894 | 98.69 | 7.79 | 1.00 |
| WT_6_2 | 4.80 | 52436258 | 51759879 | 676379 | 98.71 | 7.40 | |
| WT_12_1 | 5.04 | 54054132 | 53271826 | 782306 | 98.55 | 8.56 | 0.99 |
| WT_12_2 | 5.37 | 57646891 | 56798096 | 848795 | 98.53 | 9.29 | |
| WT_24_1 | 3.09 | 33003343 | 32498394 | 504949 | 98.47 | 5.52 | 0.80 |
| WT_24_2 | 5.38 | 53962216 | 53018166 | 944050 | 98.25 | 10.33 | |
| WT_48_1 | 6.32 | 62796702 | 61500435 | 1296267 | 97.94 | 14.18 | 0.99 |
| WT_48_2 | 5.14 | 34693098 | 34020090 | 673008 | 98.06 | 7.36 | |
| WT_72_1 | 6.43 | 63625946 | 62530758 | 1095188 | 98.28 | 11.98 | 0.99 |
| WT_72_2 | 5.20 | 51866662 | 50779793 | 1086869 | 97.90 | 11.89 | |
| WT_96_1 | 6.33 | 63255801 | 62039033 | 1216768 | 98.08 | 13.31 | 0.99 |
| WT_96_2 | 6.11 | 61187496 | 60080963 | 1106533 | 98.19 | 12.11 | |

DOI: https://doi.org/10.7554/eLife.50374.003
The following source data is available for  Table 1:
Source data 1. Number of reads for each ORF of *Myxococcus xanthus* at 0, 6, 12, 24, 48, 72 and 96 hr of development.
DOI: https://doi.org/10.7554/eLife.50374.004
Source data 2. RPKM values of the developmental time course and correlation scores.
RPKM values reported here were calculated from the total number of non-tRNA/rRNA containing reads. The old (MXAN_) locus tags, new gene identifiers (MXAN_RS), gene name and predicted functions or pathways in which they have been previously implicated are included. The number of missing data points and fold change were used as criteria for the developmental gene analysis. DG or reason that genes were not included in the DGs is indicated.
DOI: https://doi.org/10.7554/eLife.50374.005

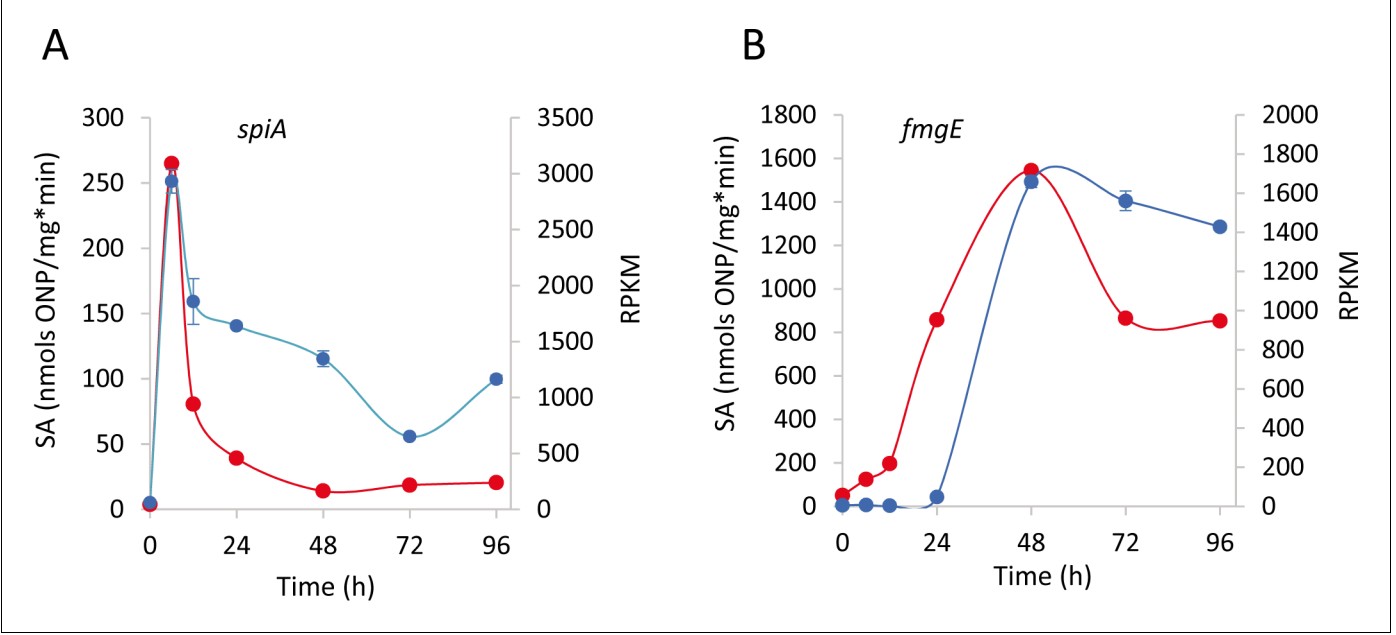

**Figure 2.** Validation of the RNA-Seq transcription patterns for genes *spiA* (MXAN_RS20760) (**A**) and *fmgE* (MXAN_RS16790) (**B**). β-galactosidase specific activity (SA) of the strains harboring *lacZ* fusions to the respective genes (blue lines) compared to RNA-Seq RPKM values (red lines) at each developmental time point (h). Error bars indicate standard deviations for β-galactosidase specific activity determination.

DOI: https://doi.org/10.7554/eLife.50374.006

The following source data is available for figure 2:

**Source data 1.** Comparison of expression profiles of several developmental genes described in the literature and included in the DGs with the RPKM profiles from this study.

DOI: https://doi.org/10.7554/eLife.50374.007

unique gene profiles (*Figure 3A*, *Figure 3—figure supplement 1*, and *Figure 3—source data 1*). Clusters were organized with peak expression profiles corresponding to progression through the developmental program (*Figure 3A*). The relative expression profiles presented here in the heat maps are a $\log_2$ normalized RPKM value relative to the mean of the entire RNA-Seq trajectory for a given gene. This relative expression profile is $\log_2(\text{RPKMi time-point x/RPKMi average})$, where i is a given gene, x is a given time-point, and RPKM average is the average RPKM value of all time-points. Although the sensitivity of the RNA-Seq technology allows detection of genes with low expression levels, some genes were excluded in this analysis after removing those with less than 50 reads. This is especially important for genes that are expressed during growth at levels not detectable with this technology. Only 13 genes were found in this situation (*Figure 3—source data 2*), which have not been included in the analyses shown below.

DGs 1 and 2 contained genes that were immediately down-regulated relative to growth conditions, DGs 3–5 correlated with the aggregation phase, DGs 6–7 correlated with the transition from aggregation to sporulation, and DGs 8–10 correlated with the sporulation phase (*Figure 3A*). The number of genes attributed to each cluster ranged from 5.5% (DG3) to 14.1% (DG8) (*Figure 3—figure supplement 2*). Similar proportions of genes exhibited peak expressions in growth (21.8%), aggregation (23.8%), and transition from aggregation to sporulation (18.8%) phases. The final phase accounted for peak expressions of 35.6% of genes, consistent with the significant morphological and physiological rewiring that must occur during spore differentiation and preparation for extended quiescence.

When the number of genes by RPKM values that account for 50% of all mRNA expression at each time point was compared, it was observed that this number was low (~300) during growth, and at 48 hr it steadily increases to 762 by the end of sporulation (*Figure 3—figure supplement 3*). This suggests that the transcriptome is becoming distributed across a broader number of genes during development.

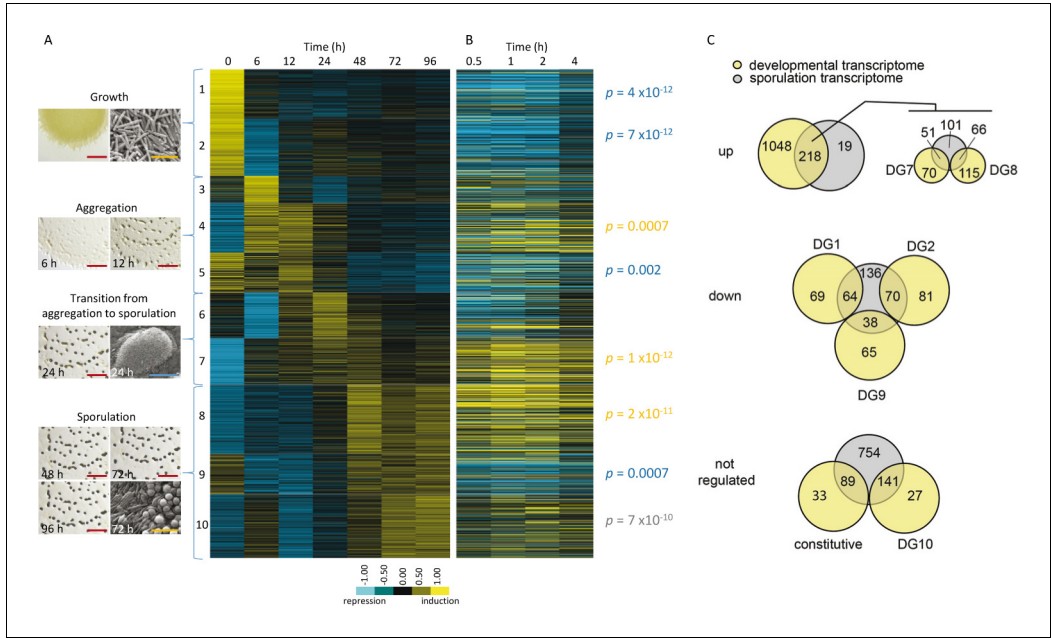

**Figure 3.** The relative expression profiles of *M. xanthus* genes observed during the developmental program compared to those previously observed during chemical-induction of sporulation. (**A**) Relative expression profiles of significantly regulated genes at the indicated hours after induction of starvation. Genes were clustered into 10 developmental groups based on the time of peak expression and then organized according to the temporal progression of development. Developmental group number and the phase of the developmental program (with photographs of aggregates under the dissecting microscope and cells under the scanning electron microscope) are indicated to the left of the heat map. In the photographs, red, blue and yellow bars represent 2 mm, 100 μm, and 5 μm, respectively. (**B**) Relative expression levels of the genes in panel A during the indicated hours after chemical-induction of sporulation (***Müller et al., 2010***). The position of individual genes in panel B is matched to panel A. Relative expression levels for panels A and B are indicated by color code according to the legend. DGs significantly represented in up-, down-, or not-regulated sporulation gene sets are indicated by the probability values in yellow, blue or gray, respectively (**C**) Comparison of up-, down-, or not-regulated starvation-induced development and chemical-induced sporulation gene tallies. Tally of glycerol-induced sporulation genes up- (top), down- (middle) or not-regulated (bottom) that are significantly enriched in the indicated DGs.
DOI: https://doi.org/10.7554/eLife.50374.008

The following source data and figure supplements are available for figure 3:

**Source data 1.** RPKM values for developmentally controlled genes and distribution of the genes in the ten developmental groups.
DOI: https://doi.org/10.7554/eLife.50374.013

**Source data 2.** Genes with no reads at time 0 hr.
DOI: https://doi.org/10.7554/eLife.50374.014

**Source data 3.** Previously reported developmental genes and identification in the 10 developmental groups shown in ***Figure 3A***.
DOI: https://doi.org/10.7554/eLife.50374.015

**Source data 4.** Genes involved in development that are not included in the DGs shown in ***Figure 3A*** and reasons for their exclusion.
DOI: https://doi.org/10.7554/eLife.50374.016

**Source data 5.** Genes included for comparison of developmental and sporulation transcriptomes (***Figure 3B***).
DOI: https://doi.org/10.7554/eLife.50374.017

**Source data 6.** Tally of enrichment of sporulation transcriptome genes (***Müller et al., 2010***) found in each of the DGs (***Figure 3B***).
DOI: https://doi.org/10.7554/eLife.50374.018

**Source data 7.** Tally of genes observed in both the developmental (this study) and sporulation [***Müller et al., 2010*** BMC Genomics 11:264] transcriptomes as supporting data for ***Figure 3C***.
DOI: https://doi.org/10.7554/eLife.50374.019

*Figure 3 continued on next page*

*Figure 3 continued*

**Figure supplement 1.** Relative expression profiles of significantly regulated genes at the indicated hours after induction of starvation.

DOI: https://doi.org/10.7554/eLife.50374.009

**Figure supplement 2.** Tally of reliable genes.

DOI: https://doi.org/10.7554/eLife.50374.010

**Figure supplement 3.** Number of genes composing 50% of the transcriptome throughout the developmental program.

DOI: https://doi.org/10.7554/eLife.50374.011

**Figure supplement 4.** A: Relationship between *M. xanthus* developmental phases, developmental groups, and a known functional interaction network.

DOI: https://doi.org/10.7554/eLife.50374.012

As a first step in analysis of the DGs, genes that have been previously described to affect *M. xanthus* development were identified. Of the ~280 characterized genes, 167 were included in the DGs. Most showed peak expression patterns at a time point that matched with the developmental phase where they have been reported to function, some of which have a well-defined role on development (*Figure 3—source data 3* and *Figure 3—figure supplement 4*). A notable exception to this are genes that are thought to function in very early stages during development, but are included in DG10. Examples of these genes are *sasS*, which is involved in A signaling (*Yang and Kaplan, 1997*); *csgA*, which encodes the C signal (*Shimkets and Rafiee, 1990*); and *romR*, which is involved in polarity control of motility (*Leonardy et al., 2007*). However, DG10 genes also showed high expression at 6 hr, consistent with early activities and raising the possibility of additional functions at the latest stage of development. Finally, not all known developmental genes appeared in the 10 DGs defined here (*Figure 3—source data 4*). Of the 108 missing genes, 28 genes encoded TCSs and 20 serine/threonine protein kinases (STPKs).

## Genes identified through chemical induction of sporulation mainly map to DG7 and DG8

A core sporulation transcriptome was previously defined using an artificial method for inducing spore differentiation (*Müller et al., 2010*). Using this method, myxospores can be induced in cells growing in rich broth culture by addition of chemicals such as 0.5 M glycerol (*Dworkin and Gibson, 1964*). Chemical-induced sporulation bypasses the requirement for starvation, motility (aggregation), and alternate cell fates (*Higgs et al., 2014*). Comparison of these two transcriptome sets determined that 1388 genes passed quality criteria in both transcriptome studies (*Figure 3B, Figure 3—source datas 5* and *6, Figure 3C*, and *Figure 3—source data 7*). 92% (218/237) of genes significantly up-regulated (>2 fold) in the sporulation transcriptome were also significantly up-regulated in the developmental transcriptome. These genes were significantly over-represented in DGs 7 and 8, with low probability ($p$) that this association is due to random chance ($p=1\times10^{-12}$ and $p=2\times10^{-11}$, respectively). DGs 7 and 8 peak expression at the transition to sporulation and sporulation phases, respectively. Co-regulated genes in these groups include those predicted to be involved in generation of sugar precursors for spore coat (see below). Likewise, genetic loci involved in spore coat synthesis (*exo*) and surface polysaccharide arrangement (*nfs*) (*Müller et al., 2012*; *Holkenbrink et al., 2014*) fell in DG8.

Interestingly, DGs 9 and 10, with peak expression profiles corresponding to the latest developmental time points, were not well represented in the up-regulated sporulation transcriptome. DG9 genes were significantly over-represented in the down-regulated sporulation transcriptome ($p=0007$). The DG9 cluster represents genes that were expressed during growth in rich media, down-regulated in response to starvation, and later up-regulated during the final phases of development (*Figure 3A*). Genes in this cluster appear to be involved in transport, respiration, and transcriptional regulation, and may be required during rapid growth and perhaps in the final maturation phases of sporulation. It is likely that the chemical-induced sporulation transcriptome may not have included very late induced sporulation genes as RNA was harvested until four-hours after induction. Chemical-induced spores are heat and sonication resistant at this stage, but final maturation may continue past this point. Finally, DG10 contained genes that were over-represented ($p=7\times10^{-10}$) in

the not-regulated sporulation transcriptome gene set. We speculate this pool of genes may be present in PRs, a cell fate enriched late in the developmental program and not present during chemical induction of sporulation. Genes in this DG encode several STPKs, the bEBP Nla26, and several proteins involved in secondary metabolite (SM) biosynthesis (see below).

Good correlation was observed in genes that were expressed during growth and down-regulated in both transcriptome sets, with genes from DGs 1, 2, 5, and nine being over-represented in the down-regulated sporulation transcriptome set (*Figure 3C*). Of the large pool of genes that were not significantly regulated in the sporulation transcriptome (754), a relatively small number (12%) were also not significantly up- or down-regulated in the developmental transcriptome. This pool of genes likely represents constitutively expressed genes that serve as good normalization markers (*Figure 3— source data 5*) and includes housekeeping genes such as the transcription termination factor Rho (MXAN_RS11995), the cell cytoskeletal protein MreB (MXAN_RS32880) (*Müller et al., 2012*; *Treuner-Lange et al., 2015*; *Fu et al., 2018*), the gliding motility and sporulation protein AglU (MXAN_RS14565) (*White and Hartzell, 2000*), and the CheA homolog DifE (MXAN_RS32400), which is required for exopolysaccharide production and social motility (*Yang et al., 2000*).

## Analysis of developmental regulated genes

All genes included in the 10 DGs were individually analyzed to find out which processes were affected during development, which may explain the different events that occur during aggregation and sporulation. Here, we have focused on six different processes.

### A- and S-motility genes exhibit different developmental expression profiles

The two phases of the *M. xanthus* lifecycle (predatory growth and development) depend on A- and S-motility engines and their associated regulatory proteins (*Pérez et al., 2014*; *Islam and Mignot, 2015*; *Mercier and Mignot, 2016*; *Schumacher and Søgaard-Andersen, 2017*). Our data have revealed that many of the motility genes were developmentally regulated and that the expression profiles of the distinct A- and S-machinery genes clearly differed during development (*Figure 4*). A-motility genes were up-regulated during early development, while S-motility genes first decreased at 6 hr, and then returned to growth levels during aggregation (except for *pilA*), suggesting that A motility is preferentially used by cells during aggregation. During sporulation, expression of genes encoding both motility engines decreased (*Figure 4*), although some of them peaked again at this stage. The peak of some A-motility genes during sporulation is consistent with the repurposing of certain A-motility proteins to function in spore coat assembly (*Wartel et al., 2013*). At the end of development, the expression levels of the motility genes remain high. However, it should be reminded that although mature fruiting bodies are static structures, the PRs surrounding them are motile (*O'Connor and Zusman, 1991b*).

### Gene expression patterns are consistent with use of glycogen and lipid bodies as energy sources

During growth, *M. xanthus* does not appear to consume sugars as carbon or energy sources (*Watson and Dworkin, 1968*; *Bretscher and Kaiser, 1978*). Instead, pyruvate, amino acids, and lipids are efficiently utilized which directly enter the tricarboxylic acid (TCA) cycle (*Bretscher and Kaiser, 1978*). It has been long debated as to whether *M. xanthus* utilizes a fully functioning glycolytic pathway (*Curtis and Shimkets, 2008*). It was speculated that the pathway may be utilized primarily in the gluconeogenic direction to produce sugar precursors necessary for spore coat production (*Youderian et al., 1999*; *Chavira et al., 2007*; *Getsin et al., 2013*). It is unknown how these pathways contribute to energy production during development when starving cells must synthesize energy currencies (i.e. ATP) over a period of at least three days.

Analysis of the DGs revealed that most genes involved in energy generation (pyruvate dehydrogenase complex, TCA cycle and oxidative phosphorylation) were found in DG2 (*Figure 5A*). Therefore, although they are down-regulated at 6 hr, they are later up-regulated at lower levels than during growth. In contrast, many genes encoding enzymes of the glycolytic/gluconeogenic pathway were up-regulated during development, with most reaching maximum expression levels at the completion of aggregation (24–48 hr) (*Figure 5B*). This up-regulated group included genes encoding homologs of glucokinase (*glkC*) and phosphofructokinase (*pfkA*), which are specific for the glycolytic

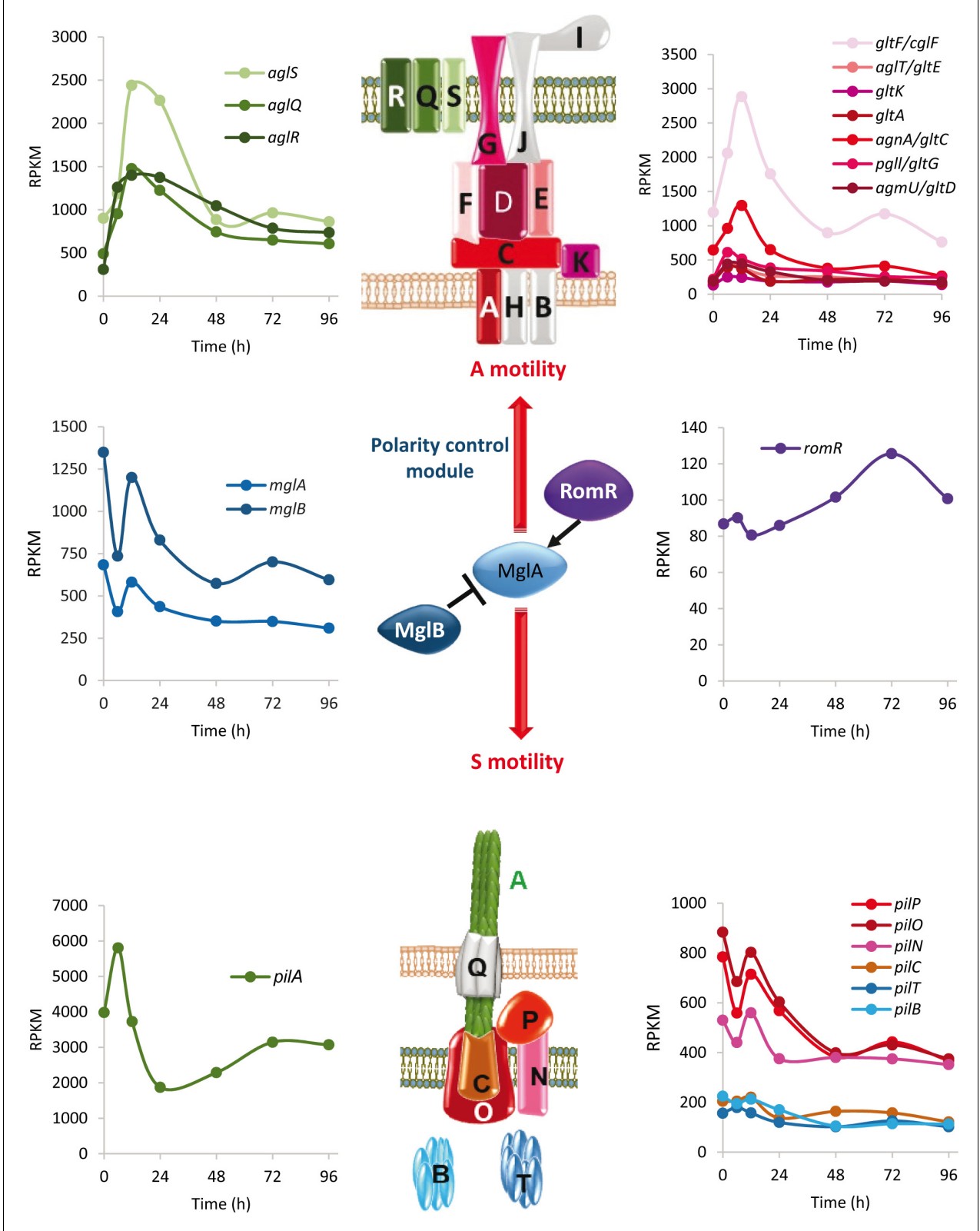

**Figure 4.** Developmental expression levels of *M. xanthus* motility proteins. Schematic representation of the focal adhesion motor complexes necessary for A motility (top), the type IV pili motor complexes necessary for S motility (bottom), and the proteins involved in controlling polarity of both engines (polarity control module; center). The developmental expression levels (RPKM) of significantly regulated motility genes at the indicated times (h) of
*Figure 4 continued on next page*

*Figure 4 continued*

development are depicted. Gene expression profiles are colored to match the proteins shown in the schematic. Proteins depicted in gray represent genes that were not included in the developmental groups.

DOI: https://doi.org/10.7554/eLife.50374.020

pathway. These observations are consistent with a transcriptional rewiring of metabolic pathways during the developmental program, perhaps to take advantage of changing carbon/energy sources. For instance, developmental up-regulation of the glycolytic pathway genes may allow developing cells to obtain energy from sugars released from cells undergoing developmental lysis or from glycogen. Glycogen accumulates during late stationary phase/early development, and then disappears prior to sporulation (*Nariya and Inouye, 2003*). Enzymes predicted to be involved in synthesis of glycogen, such as GlgC, were found in DG4, while enzymes involved in utilization of glycogen, such as trehalose synthase, GlgP, and MalQ appeared to be constitutively expressed. The observation that these competing pathways show overlapping expression profiles suggests that regulation of glycogen production/consumption is likely regulated post-transcriptionally, as has been demonstrated by phosphorylation of PfkA by Pkn4 (*Nariya and Inouye, 2003*).

Lipid bodies also accumulate in cells prior to sporulation. They mainly consist of triacylglycerides derived from membrane phospholipids as cells shorten in length, which are later used as an energy source (*Hoiczyk et al., 2009*; *Bhat et al., 2014*). However, the profiles of genes involved in both straight- and branched-chain primer synthesis and elongation of fatty acids observed in this study (*Figure 6A and B*) suggest that some level of lipid synthesis occurs during development. Moreover, genes involved in the alternative pathway to produce isovaleryl-CoA (*Bode et al., 2009*) were induced (*Figure 6A and B*). These lipids might be either incorporated into lipid bodies, be responsible for changes in lipid composition of the membranes of myxospores and/or PRs, and/or be used as precursors for SMs production. On the other hand, genes involved in lipid degradation reached maximum expression at 24–48 hr (β-oxidation) or even at later times (by other pathways) (*Figure 6C*). These fatty acid degradation enzyme profiles suggest that lipids are preferably consumed during sporulation.

These data offer an overall picture of the central metabolism during development, which reinforces the notion that macromolecules recycled from growth phase or released from lysing cells can be directly used to yield energy, but are also used to synthesize glycogen and lipids that are stored for later consumption.

## Amino acid and sugar precursors required for developmental macromolecule synthesis may be released by protein and polysaccharide turnover and gluconeogenesis

In addition to energy, starving cells need a source of sugar precursors to synthesize developmentally specific polysaccharides required for motility (*Li et al., 2003*), fruiting body encasement (*Lux and Shi, 2005*), spore coat synthesis (*Kottel et al., 1975*; *Holkenbrink et al., 2014*), and spore resistance (*McBride and Zusman, 1989*). It has been suggested that these sugars are derived from gluconeogenesis (*Youderian et al., 1999*). Consistently, our data have revealed that genes encoding enzymes specific for gluconeogenesis, such as phosphoenolpyruvate carboxykinase was in DG2, and GlpX (fructose-1,6-bisphosphatase) were present during growth and throughout development (*Table 1—source data 2*). Thus, these observations, as well as those presented above (*Figure 5B*), indicate that gluconeogenesis likely contributes to sugar precursor production at various stages during development. Moreover, the observation that four glycosyl hydrolases were specifically up-regulated during development (*Figure 7A*) suggests the cells may recycle vegetative polysaccharides or scavenge polysaccharides released from cells induced to lyse. These released free monomers could be synthesized into specific developmental polysaccharides by the series of glycosyl transferases that are also developmentally up-regulated (*Figure 7B*).

The developmental program is mainly triggered by amino acid starvation, yet developmentally specific proteins need to be newly synthesized. A source of these amino acids could be released from lysed cells and turnover of proteins that are not required during development. Consistently, 60 proteases appeared in the 10 DGs. 16 of these were specific to growth and down-regulated after 6

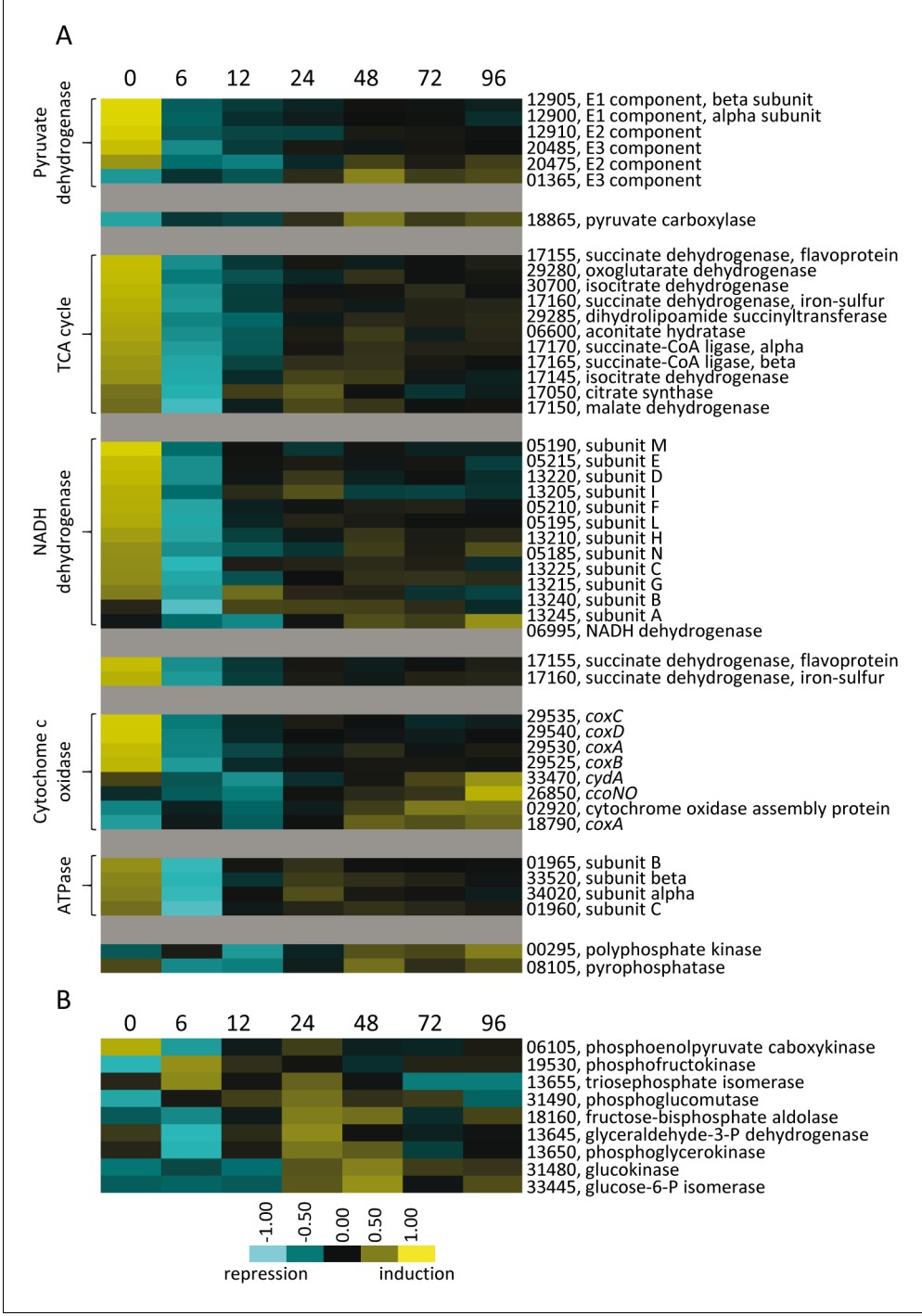

**Figure 5.** Relative developmental expression profiles of genes involved in energy generation. (**A**) Genes encoding protein homologs for the pyruvate dehydrogenase complex, TCA cycle, and oxidative phosphorylation proteins. (**B**) Genes necessary for glycolysis/gluconeogenesis. Developmental time points in hours are indicated above each panel. Relative expression levels for panels A and B are indicated by color code according to the legend. For simplicity, the MXAN_RS designation was omitted from the locus tag of each gene.

DOI: https://doi.org/10.7554/eLife.50374.021

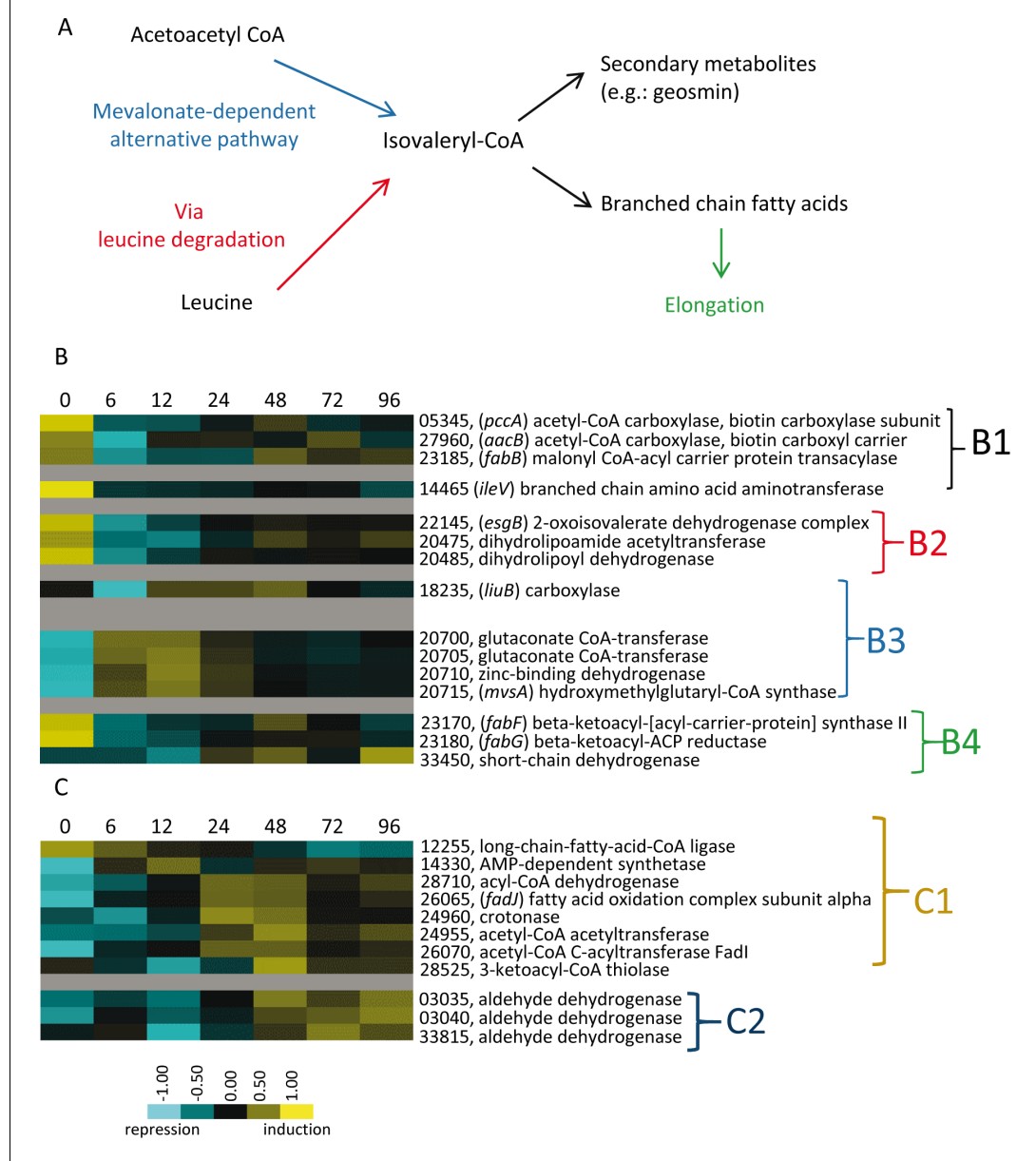

**Figure 6.** Genes involved in synthesis and degradation of lipids. (**A**) Simple representation of the *M. xanthus* branched fatty acid metabolic pathways depicting leucine degradation and alternative mevalonate-dependent routes. (**B**) Relative developmental expression profiles of the genes involved in straight-chain and branched-chain fatty acid biosynthesis as designated to the right. (**B1**) Straight-chain fatty acid primer synthesis; (**B2**) Branched-chain fatty acid primer synthesis of isovaleryl-CoA via leucine degradation (*bkd* genes); (**B3**) Branched-chain fatty acid primer synthesis of isovaleryl-CoA via the alternative pathway (mevalonate); B4: Fatty acid elongation. (**C**) Lipid degradation via β oxidation (**C1**) and other pathways (**C2**). Relative expression levels for panels B and C are indicated by color code according to the legend and developmental time points in hours are indicated above each panel. The MXAN_RS designation was omitted from the locus tag of each gene.

DOI: https://doi.org/10.7554/eLife.50374.022

hr of starvation, while the rest were sequentially up-regulated at different time points (*Figure 7C*). All of these proteases likely release amino acids, but some of them may additionally function in regulatory processes, as has been reported for the protease PopC (DG7), which is thought to function in C-signal generation by cleavage of CsgA p25 to generate the p17 fragment (*Lobedanz and Søgaard-Andersen, 2003*; *Rolbetzki et al., 2008*).

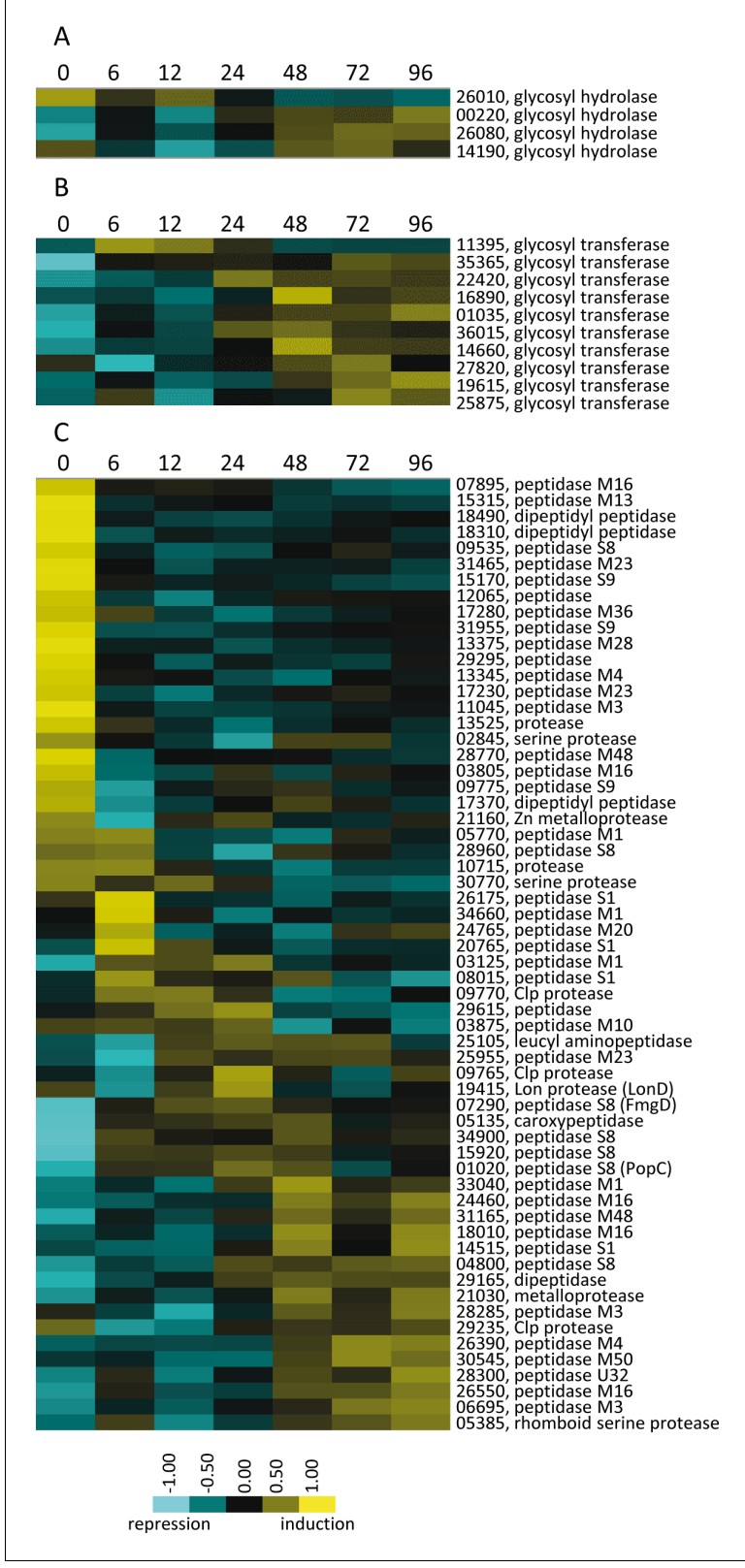

**Figure 7.** Developmental expression profiles of genes involved in production of polysaccharides and proteins. Relative expression profiles of genes predicted to be necessary for polysaccharide hydrolysis (**A**), polysaccharide synthesis (**B**), and encoding proteases and peptidases (**C**). Developmental time points in hours are indicated above

*Figure 7 continued on next page*

*Figure 7 continued*
each panel and relative expression levels are indicated by color code according to the legend at the bottom. The
MXAN_RS designation was omitted from the locus tag of each gene.
DOI: https://doi.org/10.7554/eLife.50374.023

## Genes involved in secondary metabolism are developmentally up-regulated

*M. xanthus* produces multiple SMs, some of which (i.e. myxovirescine and myxoprincomide) facilitate predation (*Xiao et al., 2011*; *Müller et al., 2016*). However, we have found that although genes responsible for their biosynthesis are expressed during growth, their expression increases during development (*Figure 8A–D*). The *M. xanthus* genome codes for 18 nonribosomal peptide synthetases (NRPS), 22 polyketide synthases (PKS), and six mixed PKS/NRPS genes located in regions predicted to be involved in SMs synthesis (*Korp et al., 2016*). Of these, 10 were developmentally up-regulated, and 9/10 were found in DGs 8, 9 and 10 (*Figure 8C–E*). Moreover, 129 genes assigned to

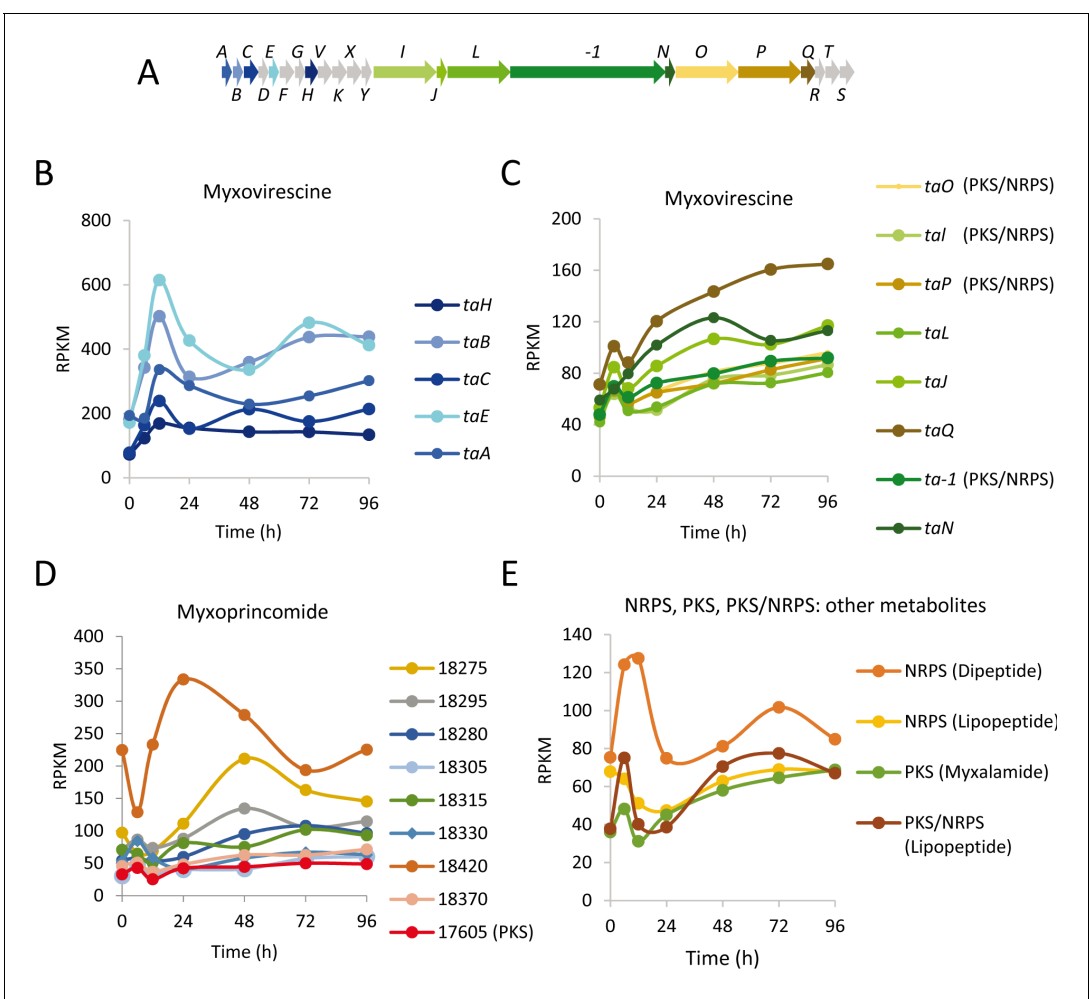

**Figure 8.** Developmental expression levels of genes involved in myxovirescine (antibiotic TA), myxoprincomide and other secondary metabolite biosynthesis. (**A**) Schematic of the myxovirescine gene cluster. Names of genes depicted here are *ta* followed by the capital letter written with each arrow. (**B**) and (**C**) Developmental expression levels (RPKM) of significantly regulated myxovirescine genes plotted against the indicated developmental time points in hours. Gene expression profiles are colored to match the genes depicted in panel A. Genes depicted in gray were not included in the developmental groups. (**D**) Expression profiles of genes involved in myxoprincomide biosynthesis. (**E**) Profiles of others NRPS, PKS and PKS/NRPS not included in panels C and D. The MXAN_RS designation was omitted from the locus tag of each gene.
DOI: https://doi.org/10.7554/eLife.50374.024

the DGs were located in those SM regions (*Korp et al., 2016*), and 51 of these were also identified in DGs 8, 9, and 10 (*Figure 3—source data 1*).

Together, these observations suggest that SMs may play previously unrecognized roles during development. For instance, they may be used to protect developing cells from other microbes in the soil, kill competitors to yield nutrients, or used as signaling molecules. In agreement with the data presented above concerning chemical-induced sporulation, an intriguing possibility is that PRs may produce SMs to defend spores inside the fruiting bodies or to release nutrients from prey to promote germination.

## Translation may be developmentally rewired

It was previously reported that the protein composition of ribosome complexes purified from growing cells versus myxospores was different (*Foster and Parish, 1973*), but this intriguing observation has not been further pursued. Data from this study suggest a role for translation regulation in control of *M. xanthus* development. 76 genes involved in translation fit the criteria for inclusion in the DGs, which represents 5.4% of the genes included in them. Most of the genes involved in translation (i.e. encoding ribosomal proteins, initiation, elongation and termination factors, ribosome maturation and modification proteins, and several aminoacyl-tRNA ligases) are down-regulated by 6 hr, likely as the result of the stringent response (*Starosta et al., 2014*). Interestingly, most of them were subsequently up-regulated to a level similar to, or even higher than that observed during growth (*Figure 9A and B*). Some genes, including several aminoacyl-tRNA ligases, were mainly expressed during sporulation (*Figure 9A and B*).

The observation that many ribosomal proteins are differentially regulated during development (*Figure 9B*) suggests that their relative ratio varies during development, yielding ribosomes with altered protein composition. It is remarkable the differences in expression profiles exhibited by duplicated genes of ribosomal proteins. *M. xanthus* encodes two paralogs for proteins S1, S4, S14, L28, and L33 (*Table 1—source data 2*). Only one of the two paralogous genes for ribosomal proteins S1 and S14 was found in the DGs (*Figure 3—source data 1*). Most notably, the two S4 paralogs were in the DGs, and exhibit similar RPKM values during growth. However, the RPKM for MXAN_RS16120 was 10-fold higher than that for MXAN_RS32955 at 48 hr of development (*Figure 9C*). Neither of the two paralogs of the 50S subunit were found in the DGs (*Table 1—source data 2*). These data confirm the previous results of *Foster and Parish (1973)*, and provide a new overall perspective on the changing composition of translational machinery during development.

While it is possible that some of the transcriptional changes observed in the translational machinery could be related to ribosomal hibernation in myxospores and/or PRs (*Yoshida and Wada, 2014*; *Harms et al., 2016*; *Gohara and Yap, 2018*), we speculate that regulation of translational machinery may play an important role in directing the developmental program. It is known that some bacteria build alternative ribosomes to improve fitness on different growth conditions by altering the core ribosomal protein stoichiometry and differential expression of paralogous ribosomal proteins (*Foster and Parish, 1973*; *Nanamiya et al., 2004*; *Hensley et al., 2012*; *Prisic et al., 2015*; *Slavov et al., 2015*). In addition, some ribosomal proteins also play extraribosomal functions, such as control of transcription or mRNA decay (*Warner and McIntosh, 2009*).

## A large interconnected regulatory network controls development

*M. xanthus* encodes a large repertoire of signaling/regulatory proteins presumed necessary to direct and coordinate its multicellular lifecycle in response to extra- and intra-cellular cues. Examples include one-component systems (OCS; transcriptional regulators that contain a sensing domain), TCS genes (sensor histidine kinase [HK] and response regulator [RR] proteins connected by phosphorelay), alternative sigma factors, and STPKs. Many of these proteins have been characterized in this bacterium, and in some cases, individual signal-transduction pathways and their exact role in controlling development have been defined (*Inouye et al., 2008*; *Schramm et al., 2012*; *Muñoz-Dorado et al., 2014*; *Rajagopalan et al., 2014*). However, the data presented here provide for the first time a view of the entire developmental cycle as an integrated system. We observed developmental up- or down-regulation of a larger number of regulatory genes than previously reported (*Muñoz-Dorado et al., 2014*; *Rajagopalan et al., 2014*). A significant number of these genes

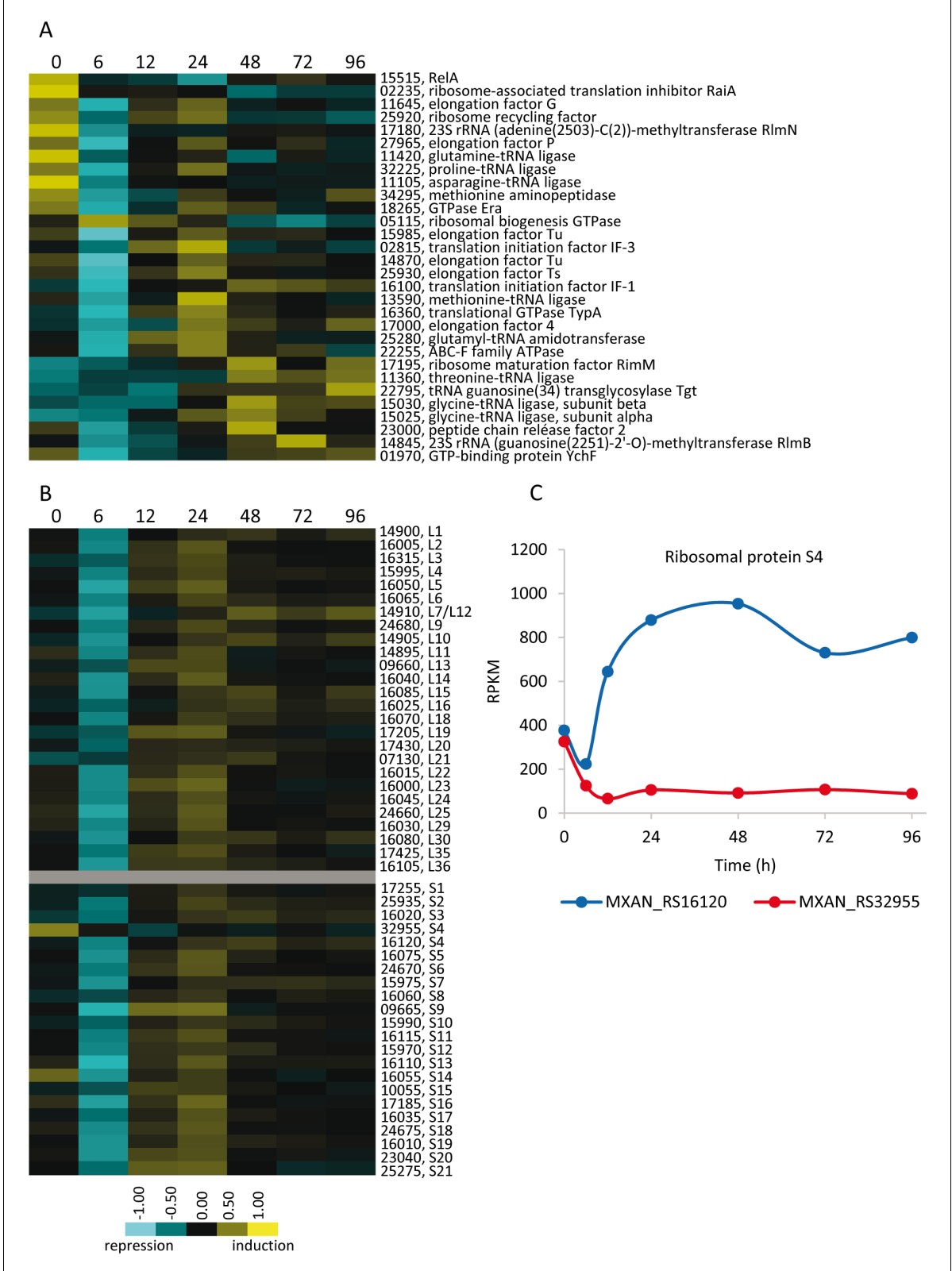

**Figure 9.** Developmental expression profiles of genes involved in protein production. (**A**) Relative expression levels of genes involved in translation or ribosome assembly. (**B**) Relative expression levels of genes encoding ribosomal proteins. Relative expression levels for panels A and B are indicated by color code according to the legend at the bottom, and developmental time points in hours are indicated above each panel. The MXAN_RS designation

*Figure 9 continued on next page*

*Figure 9 continued*
was omitted from the locus tag of each gene. (**C**) Developmental expression levels (RPKM) of the paralogous genes encoding protein S4 plotted against developmental time points in hours.
DOI: https://doi.org/10.7554/eLife.50374.025

showed expression patterns that suggested they play important roles in modulation of aggregation and/or sporulation.

21 OCS genes were developmentally regulated, 9 of which were down-regulated. The remaining were differentially up-regulated during development (*Figure 10A*), including the repressor LexA (*Campoy et al., 2003*), and the regulators SasN (*Xu et al., 1998*) and MrpC (*Sun and Shi, 2001*; *Ueki and Inouye, 2003*; *McLaughlin et al., 2018*).

Of the 272 encoded TCS genes, we found 47 included in the DGs: 23 HKs, 10 hybrid HKs (HyHKs, containing HK and RR modules in the same polypeptide), and 14 RRs. 6 TCS genes were shut down during development (*Figure 10B*). Only 16 of 47 TCS genes have been previously characterized, thus pinpointing additional candidates for further characterization. Additionally, 5 of the 14 RRs belong to the group of bEBPs (plus MXAN_RS07605, which exhibits an architecture similar to bEBPs, but contains a GAF instead of a receiver domain). bEBPs function to activate expression from sigma 54 dependent promoters (*Morett and Segovia, 1993*). Besides the bEBPs, FruA is the only other RR found in the DGs that contains a DNA-binding domain. Out of the remaining 8 RRs, five were stand-alone receiver domains (CheY-like), two contain a putative diguanylate cyclase output domain, one of which has been characterized (*Skotnicka et al., 2016*), and one is RomR, which modulates motility (*Leonardy et al., 2007*). As many of the HKs and RRs that were developmentally regulated are orphans, the results presented here may help to identify cognate pairs.

Regarding other transcription factors, nine sigma factor genes were found to be developmentally regulated in a sequential fashion (*Figure 10C*), including those encoding the major sigma factor RpoD (*Inouye, 1990*) and RpoN (sigma 54) (*Keseler and Kaiser, 1997*). The expression profiles of these two sigma factor genes clearly differ during development. While *rpoD* was down-regulated at 6 hr, up-regulated during aggregation, and then down-regulated during sporulation, *rpoN* is up-regulated throughout development (*Figure 10D*). The expression profiles of *rpoN* and bEBP genes are consistent with previous results demonstrating that a bEBP cascade is triggered upon starvation (*Giglio et al., 2011*). Additionally, *sigC*, which encodes a group II sigma factor (*Apelian and Inouye, 1993*), was identified in DG7. The remaining six are predicted to encode extracytoplasmic functions (ECF) sigma factors, including *rpoE1*, involved in motility (*Ward et al., 1998*). It is noteworthy that the genes encoding the four subunits of the core RNA polymerase are developmentally regulated with profiles similar to that observed for the ribosomal proteins (*Figure 10E*).

*M. xanthus* encodes 99 STPKs, 11 of which have been reported to be pseudokinases, because they lack at least one of the three required catalytic residues (*Muñoz-Dorado et al., 1991*; *Pérez et al., 2008*). 22 STPK genes were included in the DGs (*Figure 10F*). Interestingly, seven encode pseudokinases while 15 encode predicted active kinases (*Figure 3—source datas 1* and *3*).

Together, these observations suggest that a high number of regulators directs the developmental program of *M. xanthus*, with some acting simultaneously and others sequentially to perfectly modulate the different events that occur through development. In other model bacteria such as *Caulobacter crescentus*, more than 19% of the genes have been found to be developmentally regulated (*Laub et al., 2000*), a similar percentage to that found in *M. xanthus*, 57% of which are under direct control of the five master regulators identified in this bacterium (*Zhou et al., 2015*). In *Bacillus subtilis*, out of 4100 genes, 520 were dependent on Spo0A, but not on $\sigma^F$, while 66 were dependent upon both regulatory proteins (*Fawcett et al., 2000*). And in the case of *Streptomyces coelicolor*, 1901 genes (24% of the ORFs) exhibit differences when the substrate mycelium differentiates to a multinucleated mycelium, with a large number of transcriptional regulators involved (*Yagüe et al., 2013*). Although the number of regulators included in the DGs is larger in *M. xanthus* than in other model bacteria, it remains to be elucidated how many of them directly modulate development.

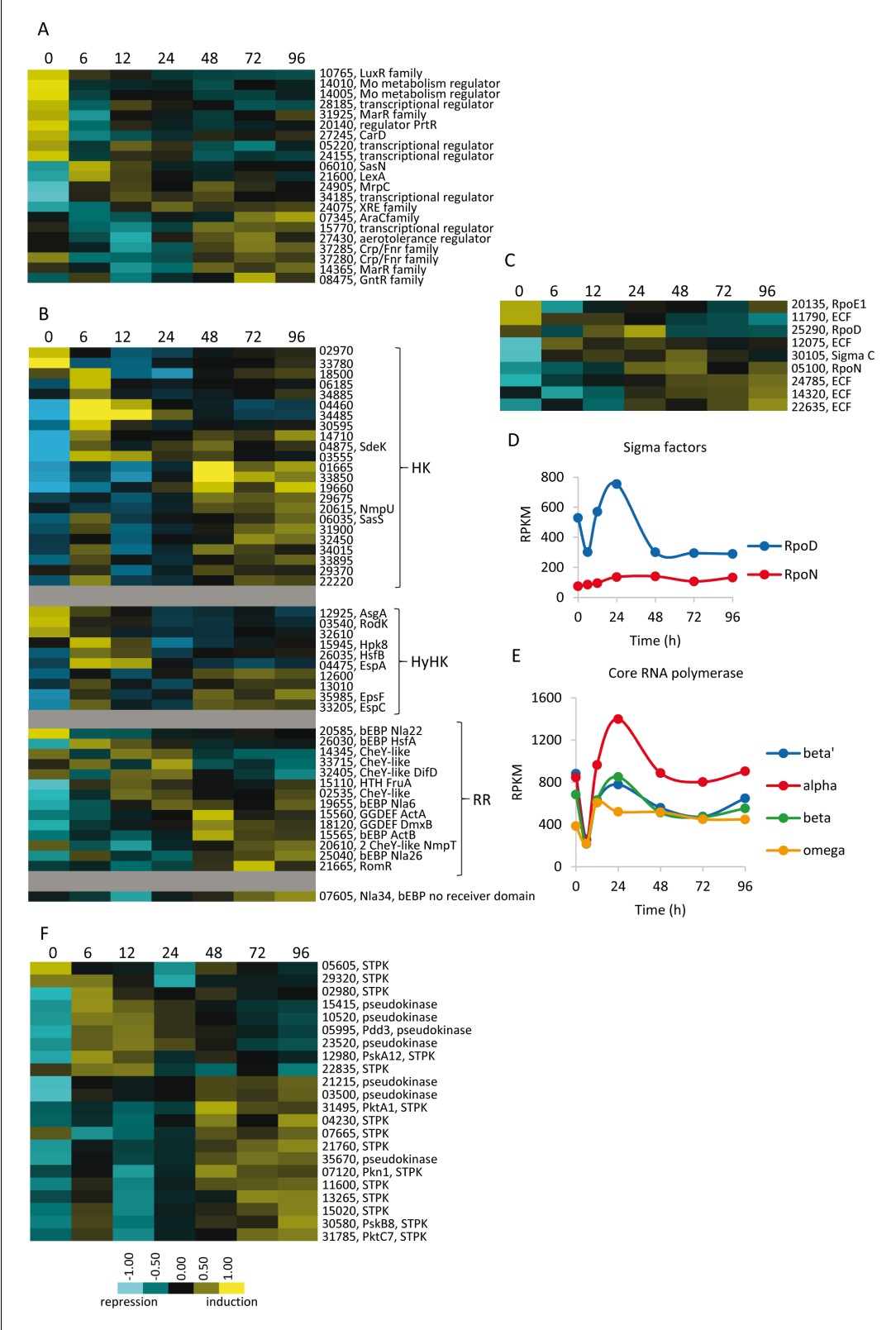

**Figure 10.** Developmental expression profiles of genes involved in transcriptional regulation and signal transduction. Relative expression levels of genes encoding one-component regulators (**A**), two-component signal transduction proteins (**B**), sigma factors (**C**), and serine/threonine protein kinases (**F**) are depicted. Relative expression levels for panels A, B, C, and F are indicated by color code according to the legend at the bottom, and developmental time points in hours are indicated above each panel. The MXAN_RS designation was omitted from the locus tag of each gene.

*Figure 10 continued on next page*

*Figure 10 continued*

Expression levels (RPKM) of genes encoding the major sigma factors (D) and the subunits of the RNA polymerase (E) plotted against developmental time points in hours.

DOI: https://doi.org/10.7554/eLife.50374.026

## Toward elucidation of a complete developmental gene-regulatory circuit

*M. xanthus* is an extraordinary bacterium with a large coding potential and a complex lifecycle. The developmental cycle consists of two consecutive events: aggregation and sporulation, during which cells segregate into three different cell fates. Moreover, this process lasts over three days and is triggered by starvation. During this extended period, starving cells must glide to build fruiting bodies and synthesize numerous macromolecules exclusive to fruiting bodies and myxospores. Consistent with this complexity, the transcriptomic analyses presented here revealed that 1415 genes are developmentally regulated with a high degree of confidence, exhibiting expression times that peak at either growth, aggregation and/or sporulation. Analysis of individual genes and the processes in which they participate has shed some light about how cells regulate the expression of motility genes to allow the cells to reach the aggregation centers, or how they rewire metabolism to both obtain energy and monomers to build new macromolecules or to synthesize SMs. Moreover, these data have also revealed that the translational and transcriptional machinery is deeply altered to modulate the different events of development, offering new insights that require to be experimentally pursued to determine the function of all these regulators.

Although the role of translation in the regulation of the developmental cycle has not yet been addressed, a large number of transcriptional regulators that function during development have been identified (*Figure 3—figure supplement 4*, and *Figure 11—source data 1*). The data presented here corroborate the results obtained by other myxobacteriologists. For instance, our analyses are in good agreement with an established model in which starvation activates four genetic regulatory networks: bEBP, Mrp, FruA, and Nla24 modules, with the latter being activated by cyclic-di-GMP (*Kroos, 2017*). However, the high number of transcriptional regulators and signaling proteins found in this study to be developmentally regulated at the mRNA level is much higher than expected. As shown in *Figure 11*, many uncharacterized regulators peak at either aggregation or sporulation (*Figure 11—source data 1*). Those that are expressed at the same time point may be interconnected to properly modulate specific events and guarantee the proper sequential expression of genes. The analysis of the expression levels of the different regulators has revealed that *fruA* and *mrpC* exhibit maximum RPKM values of 7046 and 4415, respectively, while none of the others reach 1000 (except for two sigma factors) (*Figure 11—source data 1*). This may explain why FruA and MrpC are considered master regulators of development, while many of the rest have not been identified as playing crucial roles in the lifecycle of *M. xanthus*. Undoubtedly, the developmental mRNA expression profiling presented here will act as a blueprint for the complete elucidation of the *M. xanthus* developmental regulatory program. Now that the changes in gene expression are measured, identifying the regulatory inputs of each promoter will be critical to understand the complete genetic circuitry controlling development.

## Materials and methods

**Key resources table**

| Reagent type (species) or resource | Designation | Source or reference | Identifiers | Additional information |
|---|---|---|---|---|
| Strain, strain background (*Myxococcus xanthus*) | DK1622 | *Kaiser, 1979*; *Goldman et al., 2006* | Genome: NC_008095.1 | Wild-type strain used to obtain RNA |
| Strain, strain background (*Myxococcus xanthus*) | DK4322 | *Kroos et al., 1986* | MXAN_RS20760; MXAN_4276 | *spiA*::Tn5-*lacZ*; Tn5 lac (Km$^r$) Ω4521 |

*Continued on next page*

*Continued*

| Reagent type (species) or resource | Designation | Source or reference | Identifiers | Additional information |
|---|---|---|---|---|
| Strain, strain background (*Myxococcus xanthus*) | DK4294 | *Kroos et al., 1986* | MXAN_RS16790; MXAN_ 3464 | *fmgE*::Tn5-*lacZ*; Tn5 lac (Km$^r$) Ω4406 |
| Chemical compound, drug | DNase I | Sigma-Aldrich | Cat No./ID: AMPD1 | |
| Chemical compound, drug | SuperScript III Reverse Transcriptase | Invitrogen | Cat No./ID: 18080044 | |
| Chemical compound, drug | *Escherichia coli* DNA polymerase | New England Biolabs | D1806 | |
| Chemical compound, drug | *E. coli* RNAse H | Invitrogen | Cat No./ID: 18021071 | |
| Chemical compound, drug | Proteinase K | Ambion | Cat No./ID: 25530–015 | |
| Chemical compound, drug | *E. coli* DNA ligase | New England Biolabs | Cat No./ID: M0205L | |
| Chemical compound, drug | lysozyme | Roche | Cat No./ID: 10837059001 | |
| Commercial assay or kit | RNeasy Midi Kit | Qiagen | Cat No./ID: 75142 | |
| Commercial assay or kit | RNA Protect Bacteria Reagent | Qiagen | Cat No./ID: 76506 | |
| Software, algorithm | BWA software | *Li and Durbin, 2009* | | |
| Software, algorithm | SAMtools | *Li et al., 2009* | | |
| Software, algorithm | Artemis v.16.0.0 | *Rutherford et al., 2000* | | |
| Software, algorithm | Cluster 3 Software Package | *de Hoon et al., 2004* | | |

## Preparation of cells for RNA-Seq experiment

*M. xanthus* strain DK1622 (*Kaiser, 1979*; *Goldman et al., 2006*) was used in this study. Cells were grown in CTT liquid medium (*Hodgkin and Kaiser, 1977*) at 30˚C with vigorous shaking (300 rpm) to $3.0 \times 10^8$ cells/ml (optical density at 600 nm [$OD_{600}$] of 1), and then harvested and resuspended in TM buffer (10 mM Tris-HCl [pH 7.6]; 1 mM MgSO$_4$) to a calculated density of $4.5 \times 10^9$ cells/ml ($OD_{600}$ of 15). For each time replicate, 200 µl aliquots of concentrated cell suspension were spotted onto thirteen separate CF agar plates (*Hagen et al., 1978*). Two replicates of cells were harvested from plates at 6, 12, 24, 48, 72 and 96 hr (samples WT_6, WT_12, WT_24, WT_48, WT_72 and WT_96, respectively), and the obtained pellets were transferred immediately into 0.5 ml of RNA Protect Bacteria Reagent (Qiagen). Cells were then incubated at room temperature for 5 min, harvested by centrifugation at 5000 g for 10 min (4˚C), and stored at −80˚C after removal of the supernatant. For the t = 0 samples (sample WT_0), two replicates of 30 ml of the original liquid culture ($OD_{600}$ of 1) were harvested by centrifugation as above, resuspended in RNA Protect Bacteria Reagent, and processed in the same manner.

## RNA extraction

To isolate RNA, frozen pellets were thawed and resuspended in 1 ml of 3 mg/ml lysozyme (Roche) and 0.4 mg/ml proteinase K (Ambion) prepared in TE buffer (10 mM Tris-HCl; 1 mM ethylenediaminetetraacetic acid [EDTA], pH 8.0) for cell lysis. Samples were incubated 10 min at room temperature. RNA extraction was carried out using the RNeasy Midi Kit (Qiagen) and each sample was eluted in 300 µl of RNase-free water. The concentration of RNA was measured using a NanoDrop ND-2000 spectrophotometer (NanoDrop Technologies, USA). To remove DNA, each RNA sample was supplemented with 1 unit of DNase I (from the DNA Amplification Grade Kit of Sigma) per µg of RNA and incubated at room temperature for 10 min. The reaction was stopped by adding the

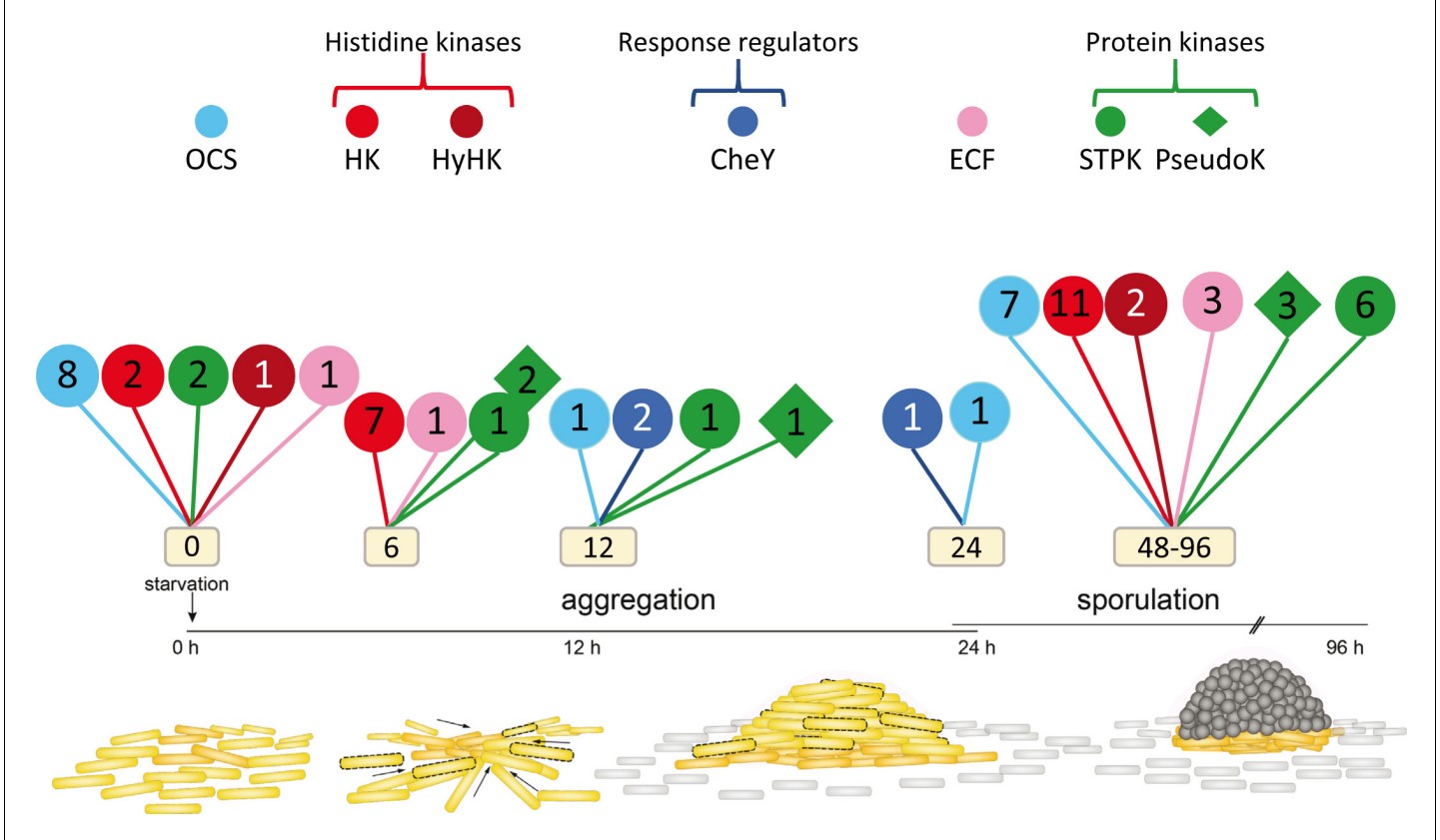

**Figure 11.** New signaling proteins that are developmentally regulated in *M. xanthus* identified in this study. Code used to distinguish among types of regulators is indicated in the upper part, where OCS indicates one-component systems; HK, histidine kinase: hyHK, hybrid histidine kinase; CheY, CheY-likeresponse regulator; RR, response regulator; ECF, ECF sigma factor; STPK, active Ser/Thr protein kinase; PseudoK, pseudokinase. Numbers inside each symbol indicate the number of each type of regulator that have not been previously identified as being developmentally regulated. Information about proteins depicted here is shown in *Figure 11—source data 1*.

DOI: https://doi.org/10.7554/eLife.50374.027

The following source data is available for figure 11:

**Source data 1.** Regulatory elements included in the developmental groups.

DOI: https://doi.org/10.7554/eLife.50374.028

stop solution included in the kit and incubating 10 min at 70°C. The obtained RNA was precipitated with 1/10 vol of 3 M sodium acetate and 3 volumes of ethanol, and resuspended in 50 µl of RNase-free water. The quality of the total RNA was verified by agarose gel electrophoresis, and the concentration was determined using NanoDrop as indicated above.

## Double stranded copy DNA synthesis

First strand DNA was synthesized using SuperScript III Reverse Transcriptase (Invitrogen) starting with 5 µg of RNA in a final reaction volume of 20 µl. In the next step, second-strand DNA was synthesized by adding 40 units of *Escherichia coli* DNA polymerase (New England Biolabs), 5 units of *E. coli* RNAse H (Invitrogen), 10 units of *E. coli* DNA ligase (New England Biolabs), 0.05 mM (final concentration) of dNTP mix, 10x second-strand buffer (New England Biolabs), and water to 150 µl. After 2 hr at 16°C, the reaction was stopped with 0.03 mM EDTA (final concentration). The obtained DNA was purified and concentrated using the DNA Clean and Concentrator Kit of Zymo Research according to manufacturer's instructions. The final product was eluted in DNA elution buffer from the kit to reach, at least, a yield of 2 µg of DNA, with a minimum concentration of 200 ng/µl.

## Sequencing and transcriptomic data analysis

The cDNA from two biological replicates of each condition (see above) was used for sequencing using the Illumina HiSeq2000 (100 bp paired-end read) sequencing platform (GATC Biotech, Germany). Sequence reads were pre-processed to remove low-quality bases. Next, reads were mapped against *M. xanthus* DK1622 ribosomal operon sequences using BWA software with the default parameters (*Li and Durbin, 2009*). Remaining reads were subsequently mapped to the genome sequence with the default parameters and using the pair-end strategy. SAMtools (*Li et al., 2009*) was used to convert resulting data into BAM format. Artemis v.16.0.0 (*Rutherford et al., 2000*) was used for the visualization of the sequence reads against the *M. xanthus* genome. Once the transcripts were mapped to the genome, the average median value for each condition was used in further analyses (*Table 1* and *Table 1—source data 1* and *2*).

## Developmental gene analysis

Genes with fewer than 50 reads in a given time point were removed from analysis. RPKM values of the remaining genes were then compared across the developmental time-points. Developmental expression was characterized by a > 2 fold change in RPKM values across the time course and a > 0.7 $R^2$ correlation coefficient of the two time course replicates. $Log_2$ fold-change calculations were performed using the Cluster 3 Software Package (*de Hoon et al., 2004*). Genes passing these criteria are presented in *Figure 3—source data 1*. Randomization of the time point RPKM values within each replicate data set yielded a false discovery rate of 3.97% based on five randomized simulations that scrambled the order of the time points across all genes in the two datasets. Genes passing the developmental expression criteria were hierarchically clustered using pearson correlation, spearman rank, euclidian distance, and kmeans clustering. By visually inspecting the clusters we found that kmeans clustering gave the best clustering of genes with similar expression profiles.

## Assay of β-galactosidase activity

For quantitative determination of β-galactosidase activity during development, strains containing *lacZ* fusions (*Figure 2*) were cultured and spotted onto CF plates as described above. Cell extracts were obtained at different times by sonication and assayed for activity as previously reported (*Moraleda-Muñoz et al., 2003*). The amount of protein in the supernatants was determined by using the Bio-Rad Protein Assay (Bio-Rad, Inc) with bovine serum albumin as a standard. Specific activity is expressed as nmol of *o*-nitrophenol (ONP) produced per min and mg of protein. The results are the average and associated standard deviation from three independent biological replicates.

## Microscopy

To observe swarm and fruiting bodies, 10 µl of cell suspension prepared as mentioned above were spotted onto CTT (for growth) or CF (for development) agar plates and incubated at 30°C. Observation was on an Olympus SZX7 dissecting microscope. For scanning electron microscopy, samples obtained from CF and CTT agar plates were fixed with glutaraldehyde vapors for 24 hr at room temperature and then postfixed in aqueous 1% osmium tetroxide for 1 hr at 4°C, washed three times in buffer, and poststained for 2 hr in buffered 0.5% uranyl acetate. Dehydration was accomplished by a graded series of ethanol. Samples were then critical-point dried and sputter coated with gold. Photographs were taken in a Zeiss DSM950 scanning electron microscope.

## Supporting data

The transcriptome sequencing data (raw-reads) was submitted to NCBI SRA under the Bioproject accession number: PRJNA493545. SRA accession numbers for each of the replicas are as follows: 0 hr: SAMN10135973 (WT_0_1-biological_replicate_1) and SAMN10135974 (WT_0_2-biological_replicate_2); 6 hr: SAMN10135975 (WT_6_1-biological_replicate_1) and SAMN10135976 (WT_6_2-biological_replicate_2); 12 hr: SAMN10135977 (WT_12_1-biological_replicate_1) and SAMN10135978 (WT_12_2-biological_replicate_2); 24 hr: SAMN10135979 (WT_24_1-biological_replicate_1) and SAMN10135980 (WT_24_2-biological_replicate_2); 48 hr: SAMN10135981 (WT_48_1-biological_replicate_1) and SAMN10135982 (WT_48_2-biological_replicate_2); 72 hr: SAMN10135983 (WT_72_1-biological_replicate_1) and SAMN10135984 (WT_72_2-biological_replicate_2); 96 hr:

SAMN10135985 (WT_96_1-biological_replicate_1) and SAMN10135986 (WT_96_2-biological_replicate_2).

## Acknowledgements

This work has been supported by the Spanish Government (grants CSD2009-00006 to JMD and BFU2016-75425-P to AMM [70% funded by FEDER]), and by NIGMS of the National Institutes of Health under award number R35GM124733 to JMS. JPT and JMD were granted with fellowships of the Salvador de Madariaga Program to stay at Wayne State University for four months. PIH was funded by a grant from NSF IOS 1651921. We also want to thank Prof. Lee Kroos for providing strains.

## Additional information

### Funding

| Funder | Grant reference number | Author |
|---|---|---|
| Spanish Government | CSD2009-00006 | Jose Munoz-Dorado |
| Spanish Government | BFU2016-75425-P | Aurelio Moraleda-Muñoz |
| National Institutes of Health | R35GM124733 | Jared M Schrader |
| National Science Foundation | IOS 1651921 | Penelope I Higgs |
| Ministerio de Educación, Cultura y Deporte | Salvador de Madariaga Program | Jose Munoz-Dorado Juana Pérez |

The funders had no role in study design, data collection and interpretation, or the decision to submit the work for publication.

### Author contributions

José Muñoz-Dorado, Juana Pérez, Substantial contributions to conception and design, acquisition of data, and analysis and interpretation of data; Drafting the article and revising it critically for important intellectual content; Final approval of the version to be published; Aurelio Moraleda-Muñoz, Francisco Javier Marcos-Torres, Francisco Javier Contreras-Moreno, Acquisition of data; Revising the article critically for important intellectual content; Final approval of the version to be published; Ana Belen Martin-Cuadrado, Jared M Schrader, Acquisition of data and analysis and interpretation of data; Revising the article critically for important intellectual content; Final approval of the version to be published; Penelope I Higgs, Analysis and interpretation of data; Drafting the article and revising it critically for important intellectual content; Final approval of the version to be published

### Author ORCIDs

José Muñoz-Dorado (iD) https://orcid.org/0000-0001-7765-5687
Jared M Schrader (iD) https://orcid.org/0000-0002-5728-5882

### Decision letter and Author response

Decision letter https://doi.org/10.7554/eLife.50374.033
Author response https://doi.org/10.7554/eLife.50374.034

## Additional files

### Supplementary files

• Transparent reporting form
DOI: https://doi.org/10.7554/eLife.50374.029

### Data availability

The transcriptome sequencing data (raw-reads) was submitted to NCBI SRA under the Bioproject accession number: PRJNA493545. SRA accession numbers for each of the replicas are as follows: 0 h: SAMN10135973 (WT_0_1-biological_replicate_1) and SAMN10135974 (WT_0_2-biological_replicate_2); 6 h: SAMN10135975 (WT_6_1-biological_replicate_1) and SAMN10135976 (WT_6_2-biological_replicate_2); 12 h: SAMN10135977 (WT_12_1-biological_replicate_1) and SAMN10135978 (WT_12_2-biological_replicate_2); 24 h: SAMN10135979 (WT_24_1-biological_replicate_1) and SAMN10135980 (WT_24_2-biological_replicate_2); 48 h: SAMN10135981 (WT_48_1-biological_replicate_1) and SAMN10135982 (WT_48_2-biological_replicate_2); 72 h: SAMN10135983 (WT_72_1-biological_replicate_1) and SAMN10135984 (WT_72_2-biological_replicate_2); 96 h: SAMN10135985 (WT_96_1-biological_replicate_1) and SAMN10135986 (WT_96_2-biological_replicate_2).

The following dataset was generated:

| Author(s) | Year | Dataset title | Dataset URL | Database and Identifier |
| --- | --- | --- | --- | --- |
| Martin-Cuadrado, Ana-Belen | 2018 | Myxococcus xanthus DK1622 transcriptome (TaxID: 34) | https://www.ncbi.nlm.nih.gov/bioproject/PRJNA493545 | NCBI BioProject, PRJNA493545 |

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
