## [Decision Letter]

[Editors’ note: a previous version of this study was rejected after peer review, but the authors submitted for reconsideration and the revised version was accepted for publication. The first decision letter after peer review is shown below.]

Thank you for submitting your work entitled "Global transcriptome analysis of the *Myxococcus xanthus* multicellular developmental program" for consideration by *eLife*. Your article has been reviewed by three peer reviewers, and the evaluation has been overseen by a Reviewing Editor and a Senior Editor. The following individuals involved in review of your submission have agreed to reveal their identity: Lotte Sogaard-Andersen (Reviewer #1); Patrick Eichenberger (Reviewer #2); Roy Welch (Reviewer #3).

Our decision has been reached after consultation between the reviewers. Based on these discussions and the individual reviews below, we regret to inform you that your work cannot be considered for publication in *eLife*. All reviewers agree that the dataset is high quality and will constitute a valuable resource for the field. However, in the present manuscript, the current data are largely used to validate previous results in developmental gene regulation and as such, do not reveal "an integrated regulatory network". New analyses could take the manuscript in this direction and ways to perform them are suggested in each of the individual reviews. Given that the modification will likely take more than the two-months period that *eLife* grants for revisions, we can only offer to reconsider a deeply modified version that would take these comments into account. Please consider that if you decide to re-submit, the manuscript will have to go through another round of reviews before it can be considered for publication.

Reviewer #1:

This manuscript describes a global analysis of gene expression changes during development in *Myxococcus xanthus* using RNAseq. During development, *M. xanthus* cells segregate into three cell types. The data presented here contains the combined changes for all three cell types and no attempts were made to distinguish between cell types. Previously, two transcriptome analyses from developmental samples based on microarrays were published (Diodati et al., Nla18, a key regulatory protein required for normal growth and development of *Myxococcus xanthus*. J Bacteriol. 188, 1733-1743 (2006) and Shi et al., 2008. Given the technological developments in the past decade to map transcriptomes, the data presented here represent a clear qualitative improvement and represent a rich and important novel resource for the community. Moreover, the data are carefully analyzed and provide a number a testable hypotheses for future studies.

Comments:

1) Introduction third paragraph: Maybe briefly mention that Diodati et al. and Shi et al. previously published transcriptome analyzes of developing cells and how your data represent are different.

2) Subsection “Transcriptome analysis of the developmental program by RNA-Seq” paragraph two: I did not find microarray analyzes in the Bath et al., 2014 reference. Please clarify.

3) In the same paragraph: It is appreciated that the authors validate the quality of the RNAseq data using the two lacZ fusions. Later in the text, when expression profiles are discussed for genes that are important for development, it would be helpful to include a reference to those papers and mention if the listed genes were shown to be up- or down-regulated. Also, many more genes than the ones listed are important for development. To further validate the RNAseq data, I was wondering if it would be worthwhile to prepare a list with genes that are known to be important for development, their expression pattern from qRT-PCR or gene fusions and the expression profile reported here.

4) Subsection “Gene expression profiles organize into 10 developmental groups”: The criteria for including genes in the analysis should be more clearly described: It is my understanding that in order for a gene to be included, it needs to have >50 reads at all seven time points. If this is correct, it seems that this would for instance exclude genes that are not expressed at T=0 or strongly downregulated genes. Please clarify.

5) Throughout the data presentations: It is not clear (to this reviewer) what is meant by "relative expression profiles". If the developmental samples are compared to the T=0 samples, then the T=0 induction ratio for the T=0 samples should be 0.00 but it is not. Please clarify precisely what is shown in the expression profiles.

6) Figure 3, 7, 8 and 9: Standard deviations cannot be derived from n<3; please remove SD from the figures.

7) Figure 8AB: The logic underlying the order in which genes are listed in these two figures is not clear. Maybe it would be a good idea to list them according to function, e.g. in 8B L subunits would be listed together and S-subunits would be listed together.

8) Subsection “A large interconnected regulatory network controls development” paragraph three: DmxB is the response regulator-diguanylate cyclase responsible for the increase in c-di-GMP during development and the dmxB gene was previously shown to be upregulated during development (Skotnicka et al., 2016). Is the dmxB gene among the upregulated response regulators?

Reviewer #2:

This is a comprehensive study of the 96-hour program of fruiting body formation in the bacterium *Myxococcus xanthus*. Gene expression during the development program was investigated by collecting RNA at 7 successive time points and performing RNA-sequencing on two biological replicates. Based on clustering analyses, genes whose expression varied during development were assigned to 10 developmental groups. The composition of each group was discussed in view of 40 years of research on the model organism. This rich dataset will be invaluable to research groups working on Myxococcus and δ-proteobacteria. It might also hold some value to researchers working on other bacteria with well-characterized developmental programs (*B. subtilis, Streptomyces*, C. crescentus), but comparison to these systems has not been explored in the current version of this paper. In addition, the following points require clarification.

Subsection “Transcriptome analysis of the developmental program by RNA-Seq”: "two independent biological replicates" and "(R2 correlation>0.98), with the exception of 24-h samples (R2 correlation=0.8)". Why only two and not three? Even though the correlation was high for most samples, the fact that there was lower correlation at one time point might have justified the addition of a third replicate. Also, "high replicate variability between the two replicate datasets (R2 correlation <0.7) were removed". A third replicate would help for that category as well.

"After removing the ribosomal sequences (about 98% of the reads)" This seems like a waste of resources. I wonder why the authors did not use a method to pull down rRNA before sequencing? If the authors had done so, they would have saved money for including a third biological replicate.

Second paragraph of subsection “Transcriptome analysis of the developmental program by RNA-Seq”: "two independent biological replicates". The authors should insist on how the use of RNA-seq extends previous knowledge acquired from microarray experiments. The impact of the present paper is lessened if the data reported are just a confirmation of previous work.

Reviewer #3:

The authors perform a global analysis of the *Myxococcus xanthus* developmental transcriptome, taking timepoints throughout the biofilm's starvation stress response past the self-organization of fruiting bodies and the differentiation of cells into spores (96 hours). They then compare these data to prior studies, both single gene and high throughput, to determine if the changes in RNA levels for different genes at different time points either match previous data or agree with contemporary functional interaction models. Their comparison serves both as data validation and as the primary source of their conclusions. Because the authors choose to largely limit their manuscript in this way, they miss several opportunities to perform a more varied and compelling set of analyses, and ultimately fails to deliver on their impact statement; the work, as presented, does not "reveal a genetic regulatory network"; with significant modification, it might.

Many interesting questions that could be addressed by these data are left unanswered. There is enough new knowledge to merit publication, but it will require additional analysis and substantial reorganization. Because the changes are significant, there is more than one way to make them, but at least three core problems need to be addressed in one way or another.

Problem 1: Analysis of the primary expression data set is not explained in sufficient detail for the reader to understand and reproduce it. Statistics were sometimes provided in the text with incomplete descriptions of analytical methods. For example, what specific methods were used to make the log2FC comparisons? How were criteria established for determining which genes fell into each DG? Were alternative methods tried before settling on these 10 DG clusters, and could alternative methods produce a different number of clusters and/or parsing of genes within them?

Proposed method to address Problem 1: Additional analyses should be performed on the primary data set to address the following questions: Would application of statistical methods, such as multiple range tests, confirm the stability of the 10 DGs, and are the differences in alternative methods statistically relevant? How much of the transcriptome is involved at each time point (i.e. what percentage of the genome is regulated, and how does that percentage change across the time points)? Are there genes that seem to be unique to each time point (i.e. are there any genes that are significantly up- or down-regulated only at one time point and may therefore be particularly important at that specific time point)? Are there other important expression patterns that can be revealed through different statistical methods, such as a functional PCA (i.e. what is the dimensionality of the time-course data)?

Problem 2: The main focus of the middle part of the manuscript (Figures 3-9) seems to be the confirmation of an existing 'consensus' interaction network, and the authors accomplish this by providing a supporting narrative. Although this narrative is interesting, the choice of genes to focus on and the alternative explanations for inconsistent data make this part of the manuscript subjective. Developmental data already exists for the majority of genes in the 'consensus' interaction network (Figure 1B); sometimes these data are from individual gene studies and sometimes they are from high-throughput studies (also, the authors should carefully qualify their claims of novelty and double check some references). This prior work can be used to generate hypotheses that can be tested using the their data set.

Proposed method to address Problem 2: The hypothesis could be something like "given the consensus functional interaction network and data from prior studies, we expect the expression patterns of this set of genes to follow the interaction network in the following way…". The hypothesis must include genes that do not meet the authors' expression cutoffs. Also, crucial conditional statements and plausible alternative explanations like the one stated in the final sentence of subsection “A‐and S‐motility genes exhibit different developmental expression profiles” can't be left to the end of a section, because they effectively negate everything that comes before it. These statements must be addressed throughout the presentation of relevant results and within the context of prior work, rather than as an afterthought.

Problem 3: Comparison of starvation-induced to glycerol-induced sporulation expression is a very interesting idea; an analysis might provide meaningful insight regarding the similarities and differences between these seemingly related events. The authors only begin to perform this analysis.

Proposed method to address Problem 3: The extent and nature of differences in gene expression between starvation-induced and glycerol-induced sporulation must be characterized and quantified. A superficial scan of Figure 2B reveals whole regions of different expression patterns. For example, what are all of the additional repressed genes in DG 9 during glycerol-induced sporulation? Could some of these genes provide insight into the differences between the two kinds of sporulation? Could some of these data be used to address the 'spore versus peripheral rod' alternative hypothesis proposed in subsection “A‐and S‐motility genes exhibit different developmental expression profiles”? Could some of these data be used to support the 'consensus' functional interaction network?

There are more minor issues involving the consistent use of abbreviations and nomenclature, claims of novelty, appropriate references, and the accuracy of concluding statements, but these can wait for the next round of revisions.

Major Points: Text

The text should be divided into three main sections: a comprehensive analysis of the gene expression data generated for this study (see point 1 above), a detailed comparison of these data to a 'consensus' functional interaction network (see point 2 above), and a detailed comparison of these data to the glycerol-induced sporulation time course (see point 3 above). All validation work (i.e. B-gal) should be put in supplementary materials, including figures that parse the expression data to support the narrative.

Major Points: Figures

The figures should be reorganized to better help the reader navigate the manuscript; at present, their structure and sequence do not match the text in a linear sequence. In the following discussion, the current figures are referred to as 'old Figure X' and proposed figures are referred to as 'new Figure X'.

New Figure 1A should be real images of *M. xanthus* development rather than a cartoon. It should include images that represent each of the time points taken so that the reader can see what development looks like under a microscope.

Old Figure 1B should be moved later in the manuscript because that is when it is discussed in the text.

New Figure 2 should have the cartoon of *M. xanthus* development running vertically on the left side of the heatmap accurately spaced to represent events, along with representative times. A timeline is also represented horizontally, but the authors use developmental stages and times interchangeably throughout the text, and so it would be helpful to have one diagram that runs along one axis showing clearly which times refer to which DGs and which developmental stages. The times should also match new Figure 1, so that the reader can easily go back and see what each stage actually looks like. Old Figure 2B should be moved later in the manuscript because the comparison to sporulation data is not yet discussed in the text.

New Figure 3 should include at least some set of analyses (see above).

Old Figures 3 through 9 should be moved to supplementary figures.

New Figure 4 should have a diagram showing the interaction of genes like one in Figure 1B. The Results section must be clear regarding which genes have transcription profiles that support the authors' hypotheses and which ones don't, and the figure should also include any genes that didn't make the expression cutoff to be included in the analysis. The interaction network diagram in old Figure 1B is designed to horizontally match to the cartoon showing development in old Figure 1A – this is a good idea for new Figure 4, but the matching should be more obvious, and should include developmental stages and DG groupings as in new Figure 2.

New Figure 5 should be the comparison between starvation induced development/sporulation vs glycerol-induced sporulation transcription profiles. Emphasis should be focused on genes whose expression profiles are similar between the two data sets, and genes whose expression profiles are different. Perhaps some of the genes from new Figure 4 could be identified as involved in development, sporulation, or both. There may also need to be a new Figure 6 providing additional statistical analysis, similar to new Figure 3.

[Editors’ note: what now follows is the decision letter after the authors submitted for further consideration.]

Thank you for submitting your article "Transcriptome dynamics of the *Myxococcus xanthus* multicellular developmental program" for consideration by *eLife*. Your article has been reviewed by two peer reviewers, and the evaluation has been overseen by a Reviewing Editor and Gisela Storz as the Senior Editor. The following individuals involved in review of your submission have agreed to reveal their identity: Lotte Sogaard-Andersen (Reviewer #2).

The reviewers have discussed the reviews with one another and the Reviewing Editor has drafted this decision to help you prepare a revised submission.

Summary:

The authors have made considerable changes in the latest revision. Most importantly, the statistical analyses involved in the parsing and evaluation of expression data are now described clearly and early in the manuscript. Of course the authors' data are not perfect but, for these kinds of large data sets, variations between experimental replicates must be expected. For example, the relatively weak correlation between replicates of the 24 hour time point does not diminish the overall impact of the manuscript, even though the authors can only speculate about an explanation. The inclusion in supplementary materials of alternate heatmaps for different numbers of Developmental Groups (Figure 2A—figure supplement 1) is also very helpful to a reader who may well use these data for the authors' primary stated purpose – as "important tools and resources for future studies." The comparison between starvation and glycerol induced spores is particularly interesting.

---

## [Author Response]

[Editors’ note: the author responses to the first round of peer review follow.]

Our decision has been reached after consultation between the reviewers. Based on these discussions and the individual reviews below, we regret to inform you that your work cannot be considered for publication in eLife. All reviewers agree that the dataset is high quality and will constitute a valuable resource for the field. However, in the present manuscript, the current data is largely used to validate previous results in developmental gene regulation and as such, it does not reveal "an integrated regulatory network". New analyses could take the manuscript in this direction and ways to perform them are suggested in each of the individual reviews. Given that the modification will likely take more than the two-months period that eLife grants for revisions, we can only offer to reconsider a deeply modified version that would take these comments into account. Please consider that if you decide to re-submit, the manuscript will have to go through another round of reviews before it can be considered for publication.

First, I would like to thank the three reviewers for their valuable comments. Undoubtedly, their suggestions have improved the manuscript. This kind of manuscript, with such a huge amount of data, are difficult to handle and focus, and I really think they have provided many inputs for improvement. Accordingly, we have done a great effort to address all their suggestions and concerns. As it can be observed in the new version, we have reorganized and rewritten most of the manuscript to make clear what it was known and what we consider are our most relevant contributions. Moreover, we have done new analyses, and accordingly, a considerable number of new figures and source data are included in the revised version. Furthermore, I would like to mention that, from our point of view, the most important part of the manuscript is to release all the raw data to the public. We consider that these data are very useful, and that they will be a great tool for many researchers. Therefore, I hope we have satisfactorily answered the reviewers’ concerns. Alternatively, I hope we have satisfactorily justified why we have not attended a few demands.

Reviewer #1:This manuscript describes a global analysis of gene expression changes during development in Myxococcus xanthus using RNAseq. During development, M. xanthus cells segregate into three cell types. The data presented here contains the combined changes for all three cell types and no attempts were made to distinguish between cell types. Previously, two transcriptome analyses from developmental samples based on microarrays were published (Diodati et al., Nla18, a key regulatory protein required for normal growth and development of Myxococcus xanthus. J Bacteriol. 188, 1733-1743 (2006) and Shi et al., 2008. Given the technological developments in the past decade to map transcriptomes, the data presented here represent a clear qualitative improvement and represent a rich and important novel resource for the community. Moreover, the data are carefully analyzed and provide a number a testable hypotheses for future studies.Comments:1) Introduction third paragraph: Maybe briefly mention that Diodati et al. and Shi et al. previously published transcriptome analyzes of developing cells and how your data represent are different.

These authors did not use microarrays, but they analyze data from other researchers to determine which genes involved in lipid metabolism are regulated during development. I think we have included this reference now in a more proper manner.

2) Subsection “Transcriptome analysis of the developmental program by RNA-Seq” paragraph two: I did not find microarray analyzes in the Bath et al., 2014 reference. Please clarify.

These authors did not use microarrays, please see response to previous point.

3) In the same paragraph: It is appreciated that the authors validate the quality of the RNAseq data using the two lacZ fusions. Later in the text, when expression profiles are discussed for genes that are important for development, it would be helpful to include a reference to those papers and mention if the listed genes were shown to be up- or down-regulated. Also, many more genes than the ones listed are important for development. To further validate the RNAseq data, I was wondering if it would be worthwhile to prepare a list with genes that are known to be important for development, their expression pattern from qRT-PCR or gene fusions and the expression profile reported here.

For simplicity, we do not mention in the text all genes that have been previously described to be important for development. All those genes, and their references, are included in the revised version in Figure 3A—source data 1 and 3. Moreover, we have added a Supplementary file 1, where we have compared the profiles obtained by microarrays, qRT-PCR and/or lacZ fusions of some studied genes (some of them shown in Figure 3A—figure supplement 4) with the profiles obtained in our study, to further validate our data.

4) Subsection “Gene expression profiles organize into 10 developmental groups”: The criteria for including genes in the analysis should be more clearly described: It is my understanding that in order for a gene to be included, it needs to have >50 reads at all seven time points. If this is correct, it seems that this would for instance exclude genes that are not expressed at T=0 or strongly downregulated genes. Please clarify.

I hope it is now clearer that we used three criteria to consider a gene as developmentally regulated: >50 reads, R2 correlation >0.7, and fold change >2 (subsection “Gene expression profiles organize into 10 developmental groups (DGs” and “Sequencing and transcriptomic data analysis”). We are aware that using filters will exclude genes that may be important for development. On the other hand, the use of these filters give a high degree of confidence for a gene to be considered as being developmentally regulated. And regarding the concern about exclusion of genes with no reads during growth, we agree with the reviewer that this deserve a special analysis. Therefore, we have analyzed genes in this situation. We have now mentioned these genes in the manuscript and a new table has also been included in the new version (Figure 3A—source data 2).

5) Throughout the data presentations: It is not clear (to this reviewer) what is meant by "relative expression profiles". If the developmental samples are compared to the T=0 samples, then the T=0 induction ratio for the T=0 samples should be 0.00 but it is not. Please clarify precisely what is shown in the expression profiles.

We have explained what “relative expression profiles” shown in the heat maps means (subsection “Developmental gene analysis”).

6) Figure 3, 7, 8 and 9: Standard deviations cannot be derived from n<3; please remove SD from the figures.

We have eliminated the error bars when only two samples were used (see all figures).

7) Figure 8AB: The logic underlying the order in which genes are listed in these two figures is not clear. Maybe it would be a good idea to list them according to function, e.g. in 8B L subunits would be listed together and S-subunits would be listed together.

We always ordered genes according to the developmental time-point when they reach maximum expression levels. Nevertheless, we have redone the panel for ribosomal proteins for two reasons: 1) differences between all genes are not very high, and 2) the reviewer is right to mention that this specific panel will be more understandable if we separate proteins of the two ribosomal subunits and we order them by their numbers (see new Figure 9).

8) Subsection “A large interconnected regulatory network controls development” paragraph three: DmxB is the response regulator-diguanylate cyclase responsible for the increase in c-di-GMP during development and the dmxB gene was previously shown to be upregulated during development (Skotnicka et al., 2016). Is the dmxB gene among the upregulated response regulators?

Sorry for this error. We have included this gene in the text and in Figure 10.

Reviewer #2:This is a comprehensive study of the 96-hour program of fruiting body formation in the bacterium Myxococcus xanthus. Gene expression during the development program was investigated by collecting RNA at 7 successive time points and performing RNA-sequencing on two biological replicates. Based on clustering analyses, genes whose expression varied during development were assigned to 10 developmental groups. The composition of each group was discussed in view of 40 years of research on the model organism. This rich dataset will be invaluable to research groups working on Myxococcus and δ-proteobacteria. It might also hold some value to researchers working on other bacteria with well-characterized developmental programs (*B. subtilis*, Streptomyces, C. crescentus), but comparison to these systems has not been explored in the current version of this paper. In addition, the following points require clarification.

We have included a brief comparison with the transcriptome of the three bacteria the reviewer mentions (subsection “A large interconnected regulatory network controls development”).

Subsection “Transcriptome analysis of the developmental program by RNA-Seq”: "two independent biological replicates" and "(R2 correlation>0.98), with the exception of 24-h samples (R2 correlation=0.8)". Why only two and not three? Even though the correlation was high for most samples, the fact that there was lower correlation at one time point might have justified the addition of a third replicate. Also, "high replicate variability between the two replicate datasets (R2 correlation <0.7) were removed". A third replicate would help for that category as well.

We do not totally agree with the reviewer about the requirement of a third replicate, and we have several reasons that justify our disagreement. One reason is that not all the *M. xanthus* cells in the population are doing the same thing at the same time point (development is not synchronic, and there are particularly differences between the center and the edge of the spot), and for this reason we think that a third replicate would only introduce variability without adding much information. With a third replicate, many genes would be excluded because they will not accomplish the criteria we have used to consider a gene as being developmentally regulated, and they could be quite interesting for further studies, because they fit these restrictive criteria in two replicates. Actually, a few hours delay in one of the samples would make this study nearly useless. Actually, we think the R2 correlation is lower at 24 h than at other time points because some cells already started the sporulation process while other did not complete aggregation, yet (in the other time points they are either aggregating or sporulating). We have added a sentence in the manuscript to mention why variability might be higher at this specific time point. In fact, I must say that we decided to sequence two replicates instead of three for these reasons. We think this manuscript has generated new data (as those related to motility, transcription and translation) that will undoubtedly contribute to this field, because it is an excellent scaffold for future studies. Moreover, although a third replicate would offer a better statistical support, two biological replicates have been widely accepted by many authors and it is operationally fine. A correlation value of 0.8 at 24 h is still quite acceptable and does not justify the necessity of a third replicate. Finally, I would like to mention that I contacted with the journal about the need of a third replicate and they agreed that this is not necessary.

"After removing the ribosomal sequences (about 98% of the reads)" This seems like a waste of resources. I wonder why the authors did not use a method to pull down rRNA before sequencing? If the authors had done so, they would have saved money for including a third biological replicate.

There is a plethora of kits based on different methodologies to remove the rRNA from the RNA-Seq library constructions. However, it is still out on the best method for ribosomal removal and several publications have shown extensive side-by-side tests of one method versus another and have declared different winners. An additional challenge is presented by the high GC presented by *M. xanthus* genome (68.89%), in which rRNA removal is less efficient. Another factor to consider is that, at the time of sequencing, since probe based rRNA removal methods are sequence specific, the availability of specific probes for Myxococcus were not available. Summarizing, at the end of the rRNA removal process, it is frequently found high amounts of rRNA (50% or more) with the putative mRNA lost in the laboratory manipulations. As this work present a time-series with two replicates, rationale indicates that the less manipulation of the samples, the better. Therefore, we chose the option to keep the rRNA. On the analysis side, ribosomal DNA content was filtered out so that differences in rRNA reads across samples do not affect alignment rates and skew subsequent normalization of the data.

Second paragraph of subsection “Transcriptome analysis of the developmental program by RNA-Seq”: "two independent biological replicates". The authors should insist on how the use of RNA-seq extends previous knowledge acquired from microarray experiments. The impact of the present paper is lessened if the data reported are just a confirmation of previous work.

We have rewritten many parts of the manuscript to clarify how our data contribute to the previous knowledge of *M. xanthus* development. Moreover, we have written a short comparison between our transcriptome work (global) and (partial) published microarrays transcriptomes (Introduction section).

Reviewer #3:The authors perform a global analysis of the Myxococcus xanthus developmental transcriptome, taking timepoints throughout the biofilm's starvation stress response past the self-organization of fruiting bodies and the differentiation of cells into spores (96 hours). They then compare these data to prior studies, both single gene and high throughput, to determine if the changes in RNA levels for different genes at different time points either match previous data or agree with contemporary functional interaction models. Their comparison serves both as data validation and as the primary source of their conclusions. Because the authors choose to largely limit their manuscript in this way, they miss several opportunities to perform a more varied and compelling set of analyses, and ultimately fails to deliver on their impact statement; the work, as presented, does not "reveal a genetic regulatory network"; with significant modification, it might.

We have rewritten many parts of the manuscript to emphasize on all the new knowledge our data offer. We have also softened our claim of finding a genetic regulatory network.

Many interesting questions that could be addressed by these data are left unanswered. There is enough new knowledge to merit publication, but it will require additional analysis and substantial reorganization. Because the changes are significant, there is more than one way to make them, but at least three core problems need to be addressed in one way or another.Problem 1: Analysis of the primary expression data set is not explained in sufficient detail for the reader to understand and reproduce it. Statistics were sometimes provided in the text with incomplete descriptions of analytical methods. For example, what specific methods were used to make the log2FC comparisons? How were criteria established for determining which genes fell into each DG? Were alternative methods tried before settling on these 10 DG clusters, and could alternative methods produce a different number of clusters and/or parsing of genes within them?

First, I would like to mention that classification of the genes in developmental group is just one of the many results we present in this manuscript. We consider that this classification was especially important to facilitate the analyses, to visualize how the transcriptome is changing upon starvation, and to validate our data with data already published, including the transcriptome of chemically induced myxospores. Therefore, definition of the 10 DGs is one of our data, but not the most relevant one. Once the groups are defined, the current paper focuses on many single gene expression profiles of developmental genes, including those that were expected to be developmentally regulated according to work already published in the last 40 years, and many of the new ones identified here, which have not been demonstrated to be important for development. Our work provides a lot of information that undoubtedly will have to be experimentally pursued to know the exact role of each gene in development. It seems that we failed in the narrative of this last part, and we have written the new version of the manuscript to show the analyses in a clearer manner to make readers understand all the novelties our data and analyses provide. Nevertheless, and going back to the definition of the 10 clusters, we have explained in more details how the 10 DGs were obtained, and why we chose 10 instead of other number (subsections “Gene expression profiles organize into 10 developmental groups (DGs)”; “Sequencing and transcriptomic data analysis”; and “Developmental gene analysis”). We have also included a new figure (Figure 3A—figure supplement 1) to show that the election of 10 groups instead of other number was the most appropriate one.

Proposed method to address Problem 1: Additional analyses should be performed on the primary data set to address the following questions: Would application of statistical methods, such as multiple range tests, confirm the stability of the 10 DGs, and are the differences in alternative methods statistically relevant?

As mentioned above, we have explained in more details how the 10 DGs were obtained, and why we chose 10 instead of other number. We have also included a new figure to show that the election of 10 groups instead of other number was the most appropriate one.

How much of the transcriptome is involved at each time point (i.e. what percentage of the genome is regulated, and how does that percentage change across the time points)?

We have included two new figure supplements to respond this question (Figure 3A—figure supplement 2 and 3).

Are there genes that seem to be unique to each time point (i.e. are there any genes that are significantly up- or down-regulated only at one time point and may therefore be particularly important at that specific time point)?

I think the response to this question may already be in the manuscript. For instance, all genes included in DG1 are only required during growth. Additionally, all genes in DG3 peak at 6 h. In other groups and time points, we can find some genes that peak at a specific time point. Nevertheless, we have mentioned in the new version of the manuscript some genes that are involved in regulation of gene expression that peak at specific time points or specific process (aggregation, sporulation or transition from one to the other) (Figure 11).

Are there other important expression patterns that can be revealed through different statistical methods, such as a functional PCA (i.e. what is the dimensionality of the time-course data)?

Although it could be interesting, according to all the responses given above we think this is out of the scope of this manuscript. I agree with the reviewer that many things can be done with the data we have obtained, and in fact, as we mention in the manuscript, our data represent a tool and resource to work in many directions in the future. I think this suggestion (and some others mentioned) is out of this scope of the manuscript we try to publish now. Even more, I encourage Roy, as well as other myxobacteriologists, to use our data once they are public to prepare other(s) manuscript(s) that will undoubtedly help to understand the *M. xanthus* developmental cycle.

Problem 2: The main focus of the middle part of the manuscript (Figures 3-9) seems to be the confirmation of an existing 'consensus' interaction network, and the authors accomplish this by providing a supporting narrative. Although this narrative is interesting, the choice of genes to focus on and the alternative explanations for inconsistent data make this part of the manuscript subjective. Developmental data already exists for the majority of genes in the 'consensus' interaction network (Figure 1B); sometimes these data are from individual gene studies and sometimes they are from high-throughput studies (also, the authors should carefully qualify their claims of novelty and double check some references). This prior work can be used to generate hypotheses that can be tested using the their data set.

We only partially agree with this this comment. In every section we have dealt with in the manuscript, we have tried to combine what it was already known with the complexity that our data provide. It is clear that we failed to clearly differentiate these two types of data, known and novel, and our novelty seems to be hidden in the manner we presented the data. For this reason, we have rewritten all this sections to make clear what it was already known, what we have found, and how everything matches to explain *M. xanthus* developmental cycle. I would also like to mention that the previous figure addressed a streamlined functional network (and new Figure 3A—figure supplement 4). This functional network diagram illustrates those genes for which known molecular mechanisms exist.

Proposed method to address Problem 2: The hypothesis could be something like "given the consensus functional interaction network and data from prior studies, we expect the expression patterns of this set of genes to follow the interaction network in the following way…". The hypothesis must include genes that do not meet the authors' expression cutoffs. Also, crucial conditional statements and plausible alternative explanations like the one stated in the final sentence of subsection “A‐and S‐motility genes exhibit different developmental expression profiles” can't be left to the end of a section, because they effectively negate everything that comes before it. These statements must be addressed throughout the presentation of relevant results and within the context of prior work, rather than as an afterthought.

I hope the new presentation of the data satisfy this reviewer.

Problem 3: Comparison of starvation-induced to glycerol-induced sporulation expression is a very interesting idea; an analysis might provide meaningful insight regarding the similarities and differences between these seemingly related events. The authors only begin to perform this analysis.

Thank you

Proposed method to address Problem 3: The extent and nature of differences in gene expression between starvation-induced and glycerol-induced sporulation must be characterized and quantified. A superficial scan of Figure 2B reveals whole regions of different expression patterns. For example, what are all of the additional repressed genes in DG 9 during glycerol-induced sporulation? Could some of these genes provide insight into the differences between the two kinds of sporulation? Could some of these data be used to address the 'spore versus peripheral rod' alternative hypothesis proposed in subsection “A‐and S‐motility genes exhibit different developmental expression profiles”? Could some of these data be used to support the 'consensus' functional interaction network?

We think this could also be interesting, but analysis here is simply a hypothesis generator, and without experimental validation, there is no direct point to it. It could also be an interesting point to pursue in the future, but not in this manuscript. We have nevertheless addressed the differences between the two transcriptome sets more precisely and pointed out which gene sets have the potential to reveal differences in spore maturation and alternate cell types (we have rewritten the comparison in the revised version).

Major Points: TextThe text should be divided into three main sections: a comprehensive analysis of the gene expression data generated for this study (see point 1 above), a detailed comparison of these data to a 'consensus' functional interaction network (see point 2 above), and a detailed comparison of these data to the glycerol-induced sporulation time course (see point 3 above). All validation work (i.e. B-gal) should be put in supplementary materials, including figures that parse the expression data to support the narrative.

We disagree with this comment. As mentioned above, these kinds of data can be handled and many different ways, and while we are sure it would generate an interesting paper doing what the reviewer proposes, we believe that the data should be presented in the manner we have opted to do it.

Major Points: FiguresThe figures should be reorganized to better help the reader navigate the manuscript; at present, their structure and sequence do not match the text in a linear sequence. In the following discussion, the current figures are referred to as 'old Figure X' and proposed figures are referred to as 'new Figure X'.

Reorganization of the text and Figures was undertaken to address this issue as is indicated in the text.

New Figure 1A should be real images of M. xanthus development rather than a cartoon. It should include images that represent each of the time points taken so that the reader can see what development looks like under a microscope.Old Figure 1B should be moved later in the manuscript because that is when it is discussed in the text.

We have decided to maintain the cartoon of Figure 1A (Figure A in the revised version). Maybe myxobacteriologists will understand microscopic figures, but a general reader can obtain a clearer information in a cartoon where many details can be drawn that do not appear in an image (for instance, cells that lyse cannot appear in a microscopic image). Nevertheless, and as microscopic images may also be helpful, we have included them in Figure 3A. As addressed above, Figure 1B (functional network) seems to have been problematic for not including ALL genes known to be involved in development. We have deleted previous Figure 1B and recreated a new figure, which is now placed later in text as Figure 3A—figure supplement 4.

New Figure 2 should have the cartoon of M. xanthus development running vertically on the left side of the heatmap accurately spaced to represent events, along with representative times. A timeline is also represented horizontally, but the authors use developmental stages and times interchangeably throughout the text, and so it would be helpful to have one diagram that runs along one axis showing clearly which times refer to which DGs and which developmental stages. The times should also match new Figure 1, so that the reader can easily go back and see what each stage actually looks like. Old Figure 2B should be moved later in the manuscript because the comparison to sporulation data is not yet discussed in the text.

This point includes several questions. Regarding the presentation of the situation of the developmental program at each time point, we think that Figure 3A (when it is clear what happens during development depicted in Figure 1)is the moment to include pictures taken under dissecting and electron microscopes. Therefore, we have included pictures about the aggregation stage of the cells and their morphology in the new Figure 3A. Regarding the proposal to move Figure 3B to later, we feel the figure would lose a lot of the information it contains in both panels. Each gene is represented at the same position in panels A and B. Therefore, it is possible to visualize what happens with each individual gene (and group) in both situations, development by starvation and spore formation by glycerol. These two panels have to be shown side by side to provide all the information we want to show. Therefore, we have maintained panels A and B in Figure 3. Moreover, as the section about comparison between both types of spores has been moved earlier in the text, no other figure is mentioned before Figure 3B.

New Figure 3 should include at least some set of analyses mentioned (see above).

We have included a new Figure (Figure 3A—figure supplement 1) to explain why we defined 10 DGs, as mentioned above.

Old Figures 3 through 9 should be moved to supplementary figures.

We respectfully disagree again in this point. We consider all the information we show in these figures to be relevant and therefore insist they cannot be supplemental. We have a huge amount of information and it can be handled in many different manners. We consider the one we have chosen is the best one. Therefore, we have maintained Figures 4-10, with some modifications in some of them.

New Figure 4 should have a diagram showing the interaction of genes like one in Figure 1B. The Results section must be clear regarding which genes have transcription profiles that support the authors' hypotheses and which ones don't, and the figure should also include any genes that didn't make the expression cutoff to be included in the analysis. The interaction network diagram in old Figure 1B is designed to horizontally match to the cartoon showing development in old Figure 1A – this is a good idea for new Figure 4, but the matching should be more obvious, and should include developmental stages and DG groupings as in new Figure 2.

We have redone Figure 1, eliminating panel B from this figure. We agree with the reviewer that this position could be premature. We have moved this panel to supplementary data as Figure 3A—figure supplement 4, to show this information closer to a new and more complex regulatory network unveiled by our data.

New Figure 5 should be the comparison between starvation induced development/sporulation vs glycerol-induced sporulation transcription profiles. Emphasis should be focused on genes whose expression profiles are similar between the two data sets, and genes whose expression profiles are different. Perhaps some of the genes from new Figure 4 could be identified as involved in development, sporulation, or both. There may also need to be a new Figure 6 providing additional statistical analysis, similar to new Figure 3.

Our feeling is that the reviewer overlooked the information we included as supplemental with the pie charts for this part (Figure 3B—source data 1, 2 and 3), since much of what he is asking for is already there, including the stat analysis. Nevertheless, we have rearranged the section and included a new panel as Figure 3C. We recommend to examining new Figure 3B—source data 1 and 2 and Figure 3C—source data 1.